# Single-molecule imaging of microRNA-mediated gene silencing in cells

Hotaka Kobayashi [1,2,3✉] & Robert H. Singer [1✉]

MicroRNAs (miRNAs) are small non-coding RNAs, which regulate the expression of thousands of genes; miRNAs silence gene expression from complementary mRNAs through translational repression and mRNA decay. For decades, the function of miRNAs has been studied primarily by ensemble methods, where a bulk collection of molecules is measured outside cells. Thus, the behavior of individual molecules during miRNA-mediated gene silencing, as well as their spatiotemporal regulation inside cells, remains mostly unknown. Here we report single-molecule methods to visualize each step of miRNA-mediated gene silencing in situ inside cells. Simultaneous visualization of single mRNAs, translation, and miRNA-binding revealed that miRNAs preferentially bind to translated mRNAs rather than untranslated mRNAs. Spatiotemporal analysis based on our methods uncovered that miRNAs bind to mRNAs immediately after nuclear export. Subsequently, miRNAs induced translational repression and mRNA decay within 30 and 60 min, respectively, after the binding to mRNAs. This methodology provides a framework for studying miRNA function at the single-molecule level with spatiotemporal information inside cells.

[1] Department of Cell Biology, Albert Einstein College of Medicine, Bronx, NY 10461, USA. [2] PRESTO, Japan Science and Technology Agency, Chiyoda-ku, Tokyo 102-0076, Japan. [3]Present address: Institute for Quantitative Biosciences, The University of Tokyo, Bunkyo-ku, Tokyo 113-0032, Japan. ✉email: hota-koba@iqb.u-tokyo.ac.jp; robert.singer@einsteinmed.edu

MicroRNAs (miRNAs) are ~22-nt small non-coding RNAs, which silence gene expression from complementary mRNAs[1–5]. Within the human genome, there are over 500 miRNA genes[6,7], which regulate the expression of thousands of mRNAs[8], thereby influencing various biological processes and diseases. Notably, miRNAs cannot work alone; they assemble with the Argonaute subfamily of proteins (AGO) into the effector complex called the RNA-induced silencing complex (RISC)[9,10]. Using miRNAs as guides, RISC generally binds to the 3′ UTR of target mRNAs[11–13], inducing translational repression, followed by mRNA decay[14–19].

After the discovery of the first miRNA in 1993[1], miRNA-mediated gene silencing has been studied for decades. However, the function of miRNAs has been monitored primarily by ensemble methods, e.g., luciferase assays, ribosome profiling, and RNA sequencing, where a bulk collection of molecules is measured ex vivo[14–19]. Thus, the behavior of individual molecules during miRNA-mediated gene silencing, as well as their spatiotemporal regulation inside cells, remains mostly unknown. Here we report a series of single-molecule methods to visualize each step of miRNA-mediated gene silencing: RISC-binding, translational repression, and mRNA decay, in situ inside cells. As our methods visualize the function of miRNAs on a cell-by-cell basis, they enable both single-molecule and single-cell analysis. These technical advantages, which overcome the limitation of canonical methods, provide novel insights into when, where, and how miRNAs work inside cells.

## Results

### Visualization of mRNA decay by miRNAs with single-molecule resolution

First, we sought to develop a method to visualize miRNA-mediated mRNA decay with single-molecule resolution. In human U2OS cells, which have been widely used for RNA imaging[20–22], miR-21 is the most abundant miRNA[23] (Supplementary Fig. 1a–c). Thus, we constructed the reporter system that recapitulates mRNA decay by miR-21 (Fig. 1a). In this reporter system, where two different mRNAs were expressed under the control of a bi-directional promoter, firefly luciferase (Fluc) mRNAs represented the internal control. SunTag mRNAs with miR-21 sites were used to monitor miRNA-mediated mRNA decay, while SunTag mRNAs with mutant sites were used as the negative control. The miR-21 sites were designed so that the seed region (miRNA nucleotides 2–8) and the 3′ supplemental region (miRNA nucleotides 13–16) formed base-pairs with them[11–13], while the mutant sites had mismatches that interrupted base-pairing (Supplementary Fig. 1d). In this method, reporter mRNAs were detected by single-molecule fluorescence in situ hybridization (smFISH) to visualize them with single-molecule sensitivity[24,25] (Fig. 1b and Supplementary Fig. 2).

Validating our method, U2OS cells expressing SunTag mRNAs with miR-21 sites showed a smaller number of mRNAs, compared with the negative control (Fig. 1c). For quantitative analysis, we performed three-dimensional (3D) fluorescence imaging, followed by single-molecule detection in 3D using the FISH-quant algorithm[26] (Supplementary Fig. 2). This analysis confirmed the reduction of mRNA stability when SunTag mRNAs have miR-21 sites (Fig. 1d, e and Supplementary Fig. 3a–c). Although ensemble methods analyze a bulk collection of mRNAs from numerous cells[14,16–19,23], our method can analyze expression levels of mRNAs on a cell-by-cell basis. In addition, unlike canonical methods, our method can count the absolute number of mRNAs (Supplementary Fig. 3a, b), thereby allowing the absolute quantification of miRNA function.

### Visualization of translational repression by miRNAs with single-molecule resolution

Second, we attempted to develop a method to visualize miRNA-mediated translational repression with single-molecule resolution. To this end, we took advantage of the technique called single-molecule imaging of nascent peptides (SINAPS)[22], where translation of reporter mRNAs can be visualized with single-molecule resolution. Based on the principle of SINAPS, we constructed the reporter mRNA that recapitulated translational repression by miR-21 (Fig. 2a). This reporter has the SunTag sequence, consisting of 24 tandem repeats of the GCN4 epitope[27], in the ORF. Through immunofluorescence (IF) with anti-GCN4 antibodies, SunTag allows us to visualize nascent peptides being translated from mRNAs with single-molecule sensitivity. To inhibit the accumulation of SunTag throughout the cytoplasm, which dramatically increases background fluorescence, the degron sequence was added to the C terminus of the ORF[28]. To monitor miRNA-mediated translational repression, miR-21 sites were inserted in the 3′ UTR. Notably, miRNAs also trigger mRNA decay (Fig. 1c), which causes a non-negligible reduction in the number of mRNAs for analysis. To overcome this problem, we added the anti-decay sequence, $A_{114}$-$N_{40}$, which protects mRNAs against deadenylation, the first step of miRNA-mediated mRNA decay[29–32], to the end of the 3′ UTR. In this method, reporter mRNAs and their translation were visualized by smFISH and IF, respectively, with single-molecule resolution (Fig. 2b and Supplementary Fig. 4).

U2OS cells expressing reporter mRNAs without miR-21 sites showed bright SunTag signals on mRNAs (Fig. 2c, left panels, white arrowheads), indicating that translation was successfully visualized. In agreement with this, SunTag signals on mRNAs almost completely disappeared upon treatment with puromycin, an inhibitor of translation (Supplementary Fig. 5a–c). We also confirmed that adding the $A_{114}$-$N_{40}$ sequence did not have a significant impact on translation (Supplementary Fig. 5d–f). Importantly, U2OS cells expressing reporter mRNAs with miR-21 sites did not show bright SunTag signals on mRNAs (Fig. 2c, right panels, black arrowheads). The reduction of translational efficiency by miR-21 was confirmed by quantitative analysis (Fig. 2d, e and Supplementary Fig. 6a, b). Together, these results indicated that our method made it possible to visualize miRNA-mediated translational repression with single-molecule resolution. When we used this method for single-cell analysis, we found that miRNAs can completely halt translation of target mRNAs within some cells, (Fig. 2e, see cells at the bottom), while some other cells looked insensitive to the silencing (Fig. 2e, see cells at the top). This variability was independent of expression levels of target mRNAs (Supplementary Fig. 6c) and of nuclear sizes (Supplementary Fig. 6d), which correlated with the cell cycle[33,34].

Canonical methods, where a bulk collection of mRNAs is analyzed, are sufficient to monitor translational repression by miRNAs[15–19,29–32]. However, even if these methods detect a 50% reduction in translational efficiency, they cannot address how miRNAs accomplished the 50% silencing inside cells; the number of translated mRNAs may be reduced to 50%, or the number of ribosomes on translated mRNAs may be reduced to 50%. Taking advantage of single-molecule methods, we addressed this issue. To identify the number of translated mRNAs, we performed 3D-colocalization analysis between mRNAs and SunTag. In this analysis, the 3D positions of mRNAs and SunTag were localized at sub-pixel resolution by 3D Gaussian fitting[26]. Subsequently, based on the colocalization with SunTag, which is determined by the 3D distance, all mRNAs were classified into "untranslated" or "translated" mRNAs (Supplementary Fig. 4). This analysis revealed that miRNAs reduced the number of translated mRNAs within cells (Fig. 2f, g). In SINAPS experiments, dim SunTag signals that do not colocalize with mRNAs (free SunTag) represent single SunTag peptides released from ribosomes[22] (Fig. 2c, arrows). On the other hand, bright SunTag signals consist of multiple SunTag peptides being translated by multiple ribosomes (Fig. 2c, white arrowheads). Thus,

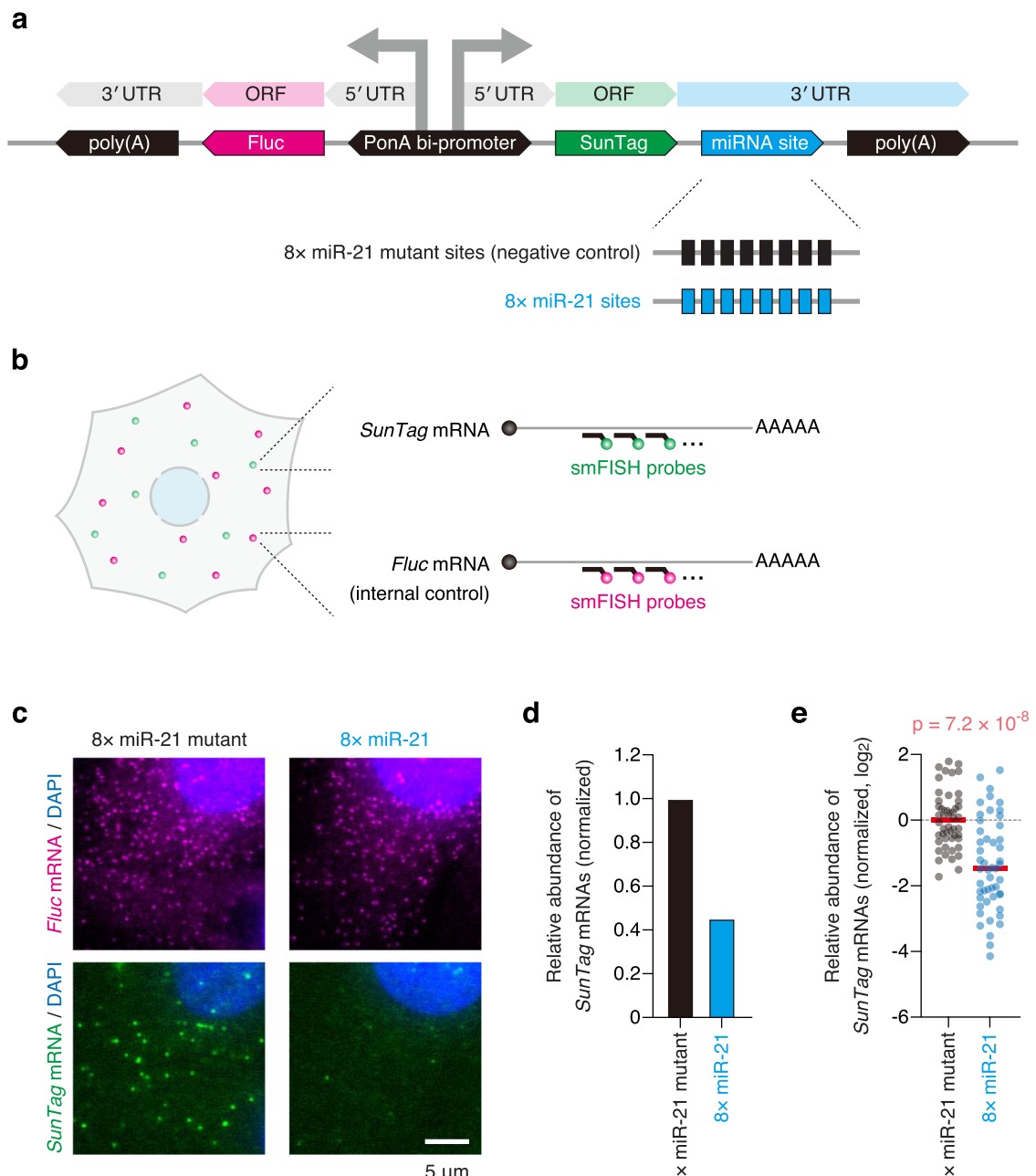

**Fig. 1 Single-molecule imaging of miRNA-mediated mRNA decay. a** Schematic of the reporter construct to recapitulate miRNA-mediated mRNA decay. PonA bi-promoter, PonA-inducible bi-directional promoter. **b** Schematic of the smFISH experiment to visualize miRNA-mediated mRNA decay at single-mRNA resolution. Green and magenta spots represent *SunTag* and *Fluc* mRNAs, respectively. **c** The images of U2OS cells expressing the 8× miR-21 mutant reporter (left) and the 8× miR-21 reporter (right). *Fluc* mRNAs (magenta, top) and *SunTag* mRNAs (green, bottom) were labeled by smFISH. Nuclei (blue) were stained by DAPI. Scale bar, 5 μm. **d, e** mRNA decay mediated by miR-21. Images were analyzed using CellProfiler and FISH-quant. Then, mRNA stability (the relative abundance of *SunTag* mRNAs to *Fluc* mRNAs) was calculated as described in Supplementary Fig. 2 (see also Methods), followed by normalization to the value of the negative control. The results of bulk analysis (**d**) and single-cell analysis (**e**) are shown. In **e**, each circle represents a single cell (*n* = 50 for each condition), while red lines represent the medians. The *p*-value of one-tailed Mann–Whitney test is shown. Source data are provided as a Source Data file.

using the fluorescence intensity of free SunTag and that of SunTag on mRNAs (Fig. 2h), the number of ribosomes on translated mRNAs could be roughly estimated[22] (Supplementary Fig. 4). This analysis revealed that miRNAs also reduced the number of ribosomes on translated mRNAs (Supplementary Fig. 6e).

**Imaging of RISC-binding with single-molecule resolution.** Third, we sought to establish a method to image RISC-binding with

single-molecule resolution. To this end, the reporter mRNA for translational repression, harboring eight miR-21 sites (Fig. 2a), was repurposed to image RISC on mRNAs. In this method, RISC was imaged by IF with anti-AGO antibodies, while reporter mRNAs were imaged by smFISH with single-molecule resolution (Fig. 3a, and Supplementary Fig. 7). Although there are four AGO proteins (AGO1-4) in humans[12], AGO2 is predominantly expressed in U2OS cells[35] (Supplementary Fig. 8a), hence we focused on AGO2 in this study. To ensure the reliability of IF with anti-AGO2 antibodies, we

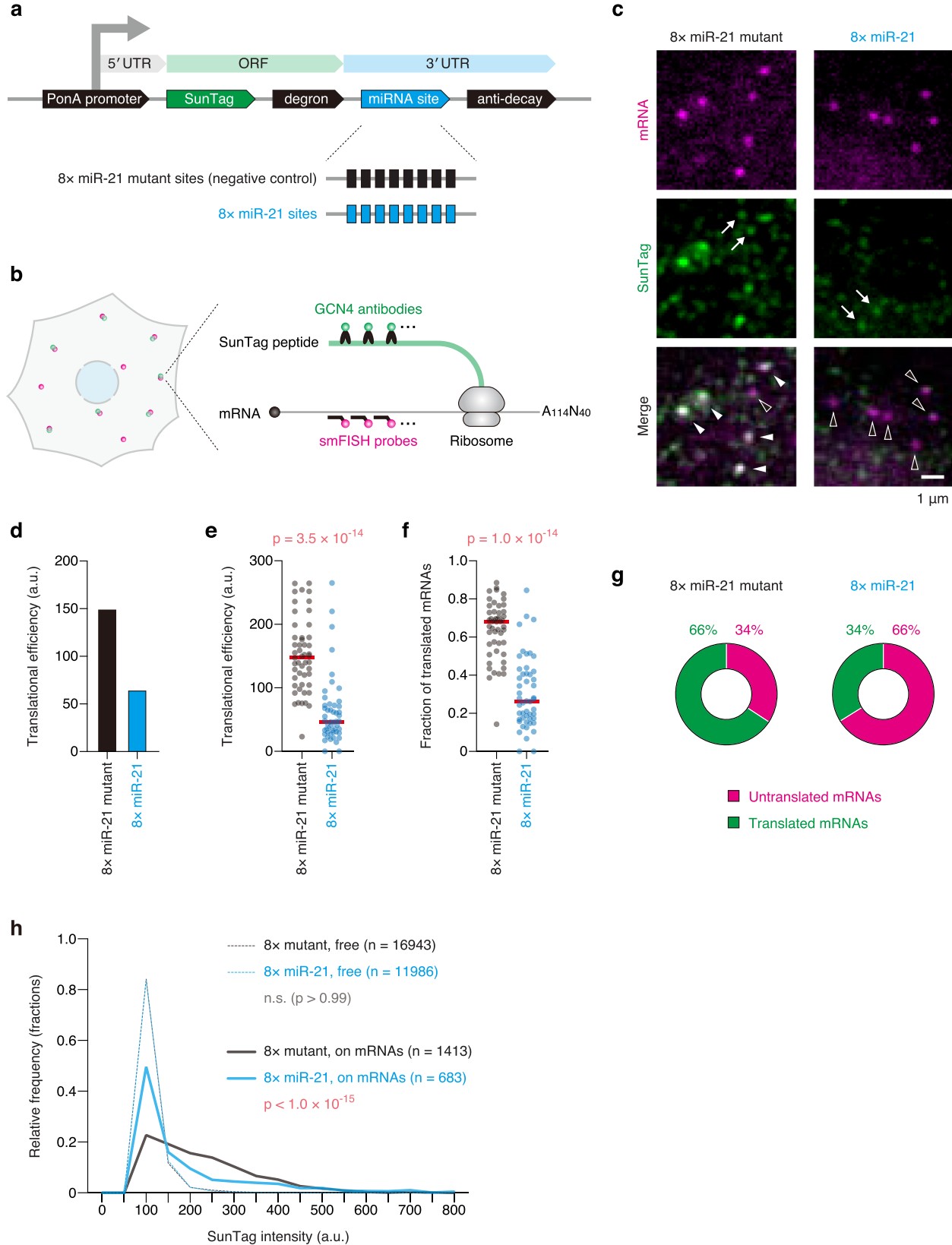

tested two distinct monoclonal antibodies (the 4G8 mouse antibody and the 11A9 rat antibody), which have already been well-characterized[36–38], and confirmed that both antibodies showed similar IF patterns (Supplementary Fig. 8b), which is consistent with previous studies looking at endogenous AGO2[39,40]. For our method, it was crucial to eliminate the unwanted RISC-binding to reporter mRNAs independent of miR-21. Thus, we carefully removed the potential miRNA sites (8mer, 7mer, and 6mer)[11] of the top 30 most abundant miRNAs from the reporter mRNAs (Supplementary Fig. 1b, c).

Given the number of AGO proteins (Supplementary Fig. 8a, ~15,000 molecules per cell) and the relative occupancy of miR-21

**Fig. 2 Single-molecule imaging of miRNA-mediated translational repression. a** Schematic of the reporter construct to recapitulate miRNA-mediated translational repression. PonA promoter, PonA-inducible promoter. **b** Schematic of the SINAPS experiment to visualize miRNA-mediated translational repression at single-mRNA resolution. Green and magenta spots represent SunTag peptides and reporter mRNAs, respectively. **c** The images of U2OS cells expressing the 8× miR-21 mutant reporter (left) and the 8× miR-21 reporter (right). Reporter mRNAs (magenta, top) and SunTag peptides (green, middle) were labeled by SINAPS. Merged images are shown at the bottom. White and black arrowheads indicate translated and untranslated mRNAs, respectively, while white arrows indicate "free" SunTag peptides. Scale bar, 1 μm. **d, e** Translational repression mediated by miR-21. Images were analyzed using CellProfiler and FISH-quant. Then, translational efficiency was calculated by division of the total SunTag intensity on mRNAs by the number of mRNAs as described in Supplementary Fig. 4 (see also Methods). The results of bulk analysis (**d**) and single-cell analysis (**e**) are shown. In **e**, each circle represents a single cell ($n = 50$ for each condition), while red lines represent the medians. The p-value of one-tailed Mann–Whitney test is shown. **f** Reduction of the fraction of translated mRNAs by miR-21. The fraction of translated mRNAs was calculated as described in Supplementary Fig. 4 (see also Methods). Each circle represents a single cell ($n = 50$ for each condition), while red lines represent the medians. The p-value of one-tailed Mann–Whitney test is shown. **g** The ratio of untranslated and translated mRNAs. All mRNAs were classified into untranslated or translated mRNAs based on 3D-colocalization analysis. **h** The histogram of SunTag intensity. The intensities of free SunTag spots (dashed lines) and of SunTag spots on mRNAs (solid lines) are shown. The p-values of Dunn's multiple comparisons test are shown. n.s., not significant. Source data are provided as a Source Data file.

(Supplementary Fig. 1c, ~25% of miRNAs) in U2OS cells, the number of RISC loaded with miR-21 is estimated to be ~4000 per cell. Thus, we minimized the expression level of reporter mRNAs (maximum, ~100; median ~40 mRNAs per cell), so that reporter mRNAs were efficiently recognized by RISC. Validating our method, U2OS cells expressing reporter mRNAs with miR-21 sites showed AGO signals on mRNAs (Fig. 3b, right panels, white arrowheads). In contrast, reporter mRNAs with mutant sites did not colocalize with AGO (Fig. 3b, left panels, black arrowheads). RISC-binding mediated by miR-21 was confirmed quantitatively by bulk analysis (Fig. 3c), single-cell analysis (Fig. 3d, e and Supplementary Fig. 8c), and single-molecule analysis (Fig. 3f, g). Notably, in some cell lines, AGO2 was found in the nucleus and in the cytoplasm at the comparable levels[41,42]. In U2OS cells, however, the majority of AGO signals was observed in the cytoplasm (Supplementary Fig. 8c), hence we focused on cytoplasmic mRNAs in our analyses. As with translational efficiency (Fig. 2e), our method highlighted the cell-to-cell variability in RISC-binding efficiency (Fig. 3d); some cells showed high RISC-binding efficiency, while RISC-binding in some other cells was comparable to the negative control. This variability was also independent of expression levels of target mRNAs (Supplementary Fig. 8d), expression levels of AGO (Supplementary Fig. 8e), and nuclear sizes (Supplementary Fig. 8f).

Although RISC-binding was successfully imaged at the single-mRNA level, our reporter mRNAs have eight miR-21 sites, so it was still unclear whether our method can detect single-RISC molecules. To investigate the sensitivity of this method to RISC, we constructed reporter mRNAs harboring a single miR-21 site or its mutant site (the negative control). When we compared AGO signals on them, there was a small difference, which was statistically significant (Supplementary Fig. 8g, h), suggesting that our method can detect at least a part of single-RISC molecules. However, given that the difference was quite small, our method would not be robust enough to detect all single-RISC molecules unequivocally. Therefore, in addition to ~30% of mRNAs that are RISC-positive (Fig. 3f, right), more mRNAs would likely be bound by RISC.

**Simultaneous visualization of single mRNAs, translation, and RISC-binding.** Since we developed a series of methods to visualize each step of miRNA-mediated gene silencing with single-molecule resolution, we next explored the relationship between these steps at the single-mRNA level. To this end, we visualized single mRNAs, translation, and RISC-binding simultaneously (Fig. 4a and Supplementary Fig. 9), using the reporter mRNA harboring miR-21 sites (Fig. 2a). Based on 3D-colocalization analyses, all mRNAs were classified into four classes: (1) RISC-negative untranslated mRNAs, (2) RISC-negative translated mRNAs, (3) RISC-positive untranslated

mRNAs, and (4) RISC-positive translated mRNAs (Fig. 4b, c). Validating our method, reporter mRNAs with miR-21 sites were classified as "RISC-positive" and "untranslated" more often than the negative control (Fig. 4c). A small portion (~10%) of reporter mRNAs without miR-21 sites were classified as "RISC-positive" (Fig. 4c, left). However, since the potential miRNA sites of major miRNAs were already removed from our reporter mRNAs (Supplementary Fig. 1b, c), this could be attributed to the non-specific colocalization between mRNAs and AGO signals by chance. In line with this, reporter mRNAs without miR-21 sites were not translationally repressed even when we focused on RISC-positive mRNAs (Supplementary Fig. 10a). It is noteworthy that a substantial amount of reporter mRNAs with miR-21 sites were classified as RISC-negative untranslated mRNAs (Fig. 4c, right). We speculate that this class would include mRNAs that are already silenced and released from RISC. Supporting this hypothesis, reporter mRNAs with miR-21 sites were translationally repressed even when we focused on RISC-negative mRNAs (Supplementary Fig. 10b). When we used the classification data for single-cell analysis, RISC-binding efficiency and translational efficiency showed a trend of negative correlation (Fig. 4d), which is consistent with the function of RISC.

Unexpectedly, when we analyzed each class of mRNAs at the single-molecule level, we found that translated mRNAs tend to be bound by RISC, compared with untranslated mRNAs (Fig. 4e and Supplementary Fig. 11a). This tendency was confirmed by quantitative analysis, where we analyzed the fluorescence intensity of AGO on untranslated mRNAs and translated mRNAs (Fig. 4f). In line with this, RISC-positive mRNAs tended to be translated, compared with RISC-negative mRNAs (Fig. 4g and Supplementary Fig. 11b). The quantitative analysis confirmed that the mRNAs bound by RISC are efficiently translated (Fig. 4h). Given that RISC does not activate translation under these conditions[43], these data indicate that RISC preferentially binds to translated mRNAs rather than untranslated mRNAs. To validate this finding, we quantified AGO signals on mRNAs in the presence or absence of a translation inhibitor, cycloheximide. This analysis confirmed that RISC-binding to reporter mRNAs was attenuated by the treatment with cycloheximide (Supplementary Fig. 11c, d). As cycloheximide immobilizes ribosomes on mRNAs, cycloheximide treatment did not change SunTag signals on mRNAs (Supplementary Fig. 11e). The results of single-mRNA analysis (Fig. 4e–h) and single-cell analysis (Fig. 4d) may appear to be contradictory at first glance. Although RISC showed a preference for translated mRNAs at the single-mRNA level, RISC is the molecule that represses translation. We speculate, therefore, cells with high or low RISC-binding efficiency corresponded to cells with low or high translational efficiency at the single-cell level, respectively.

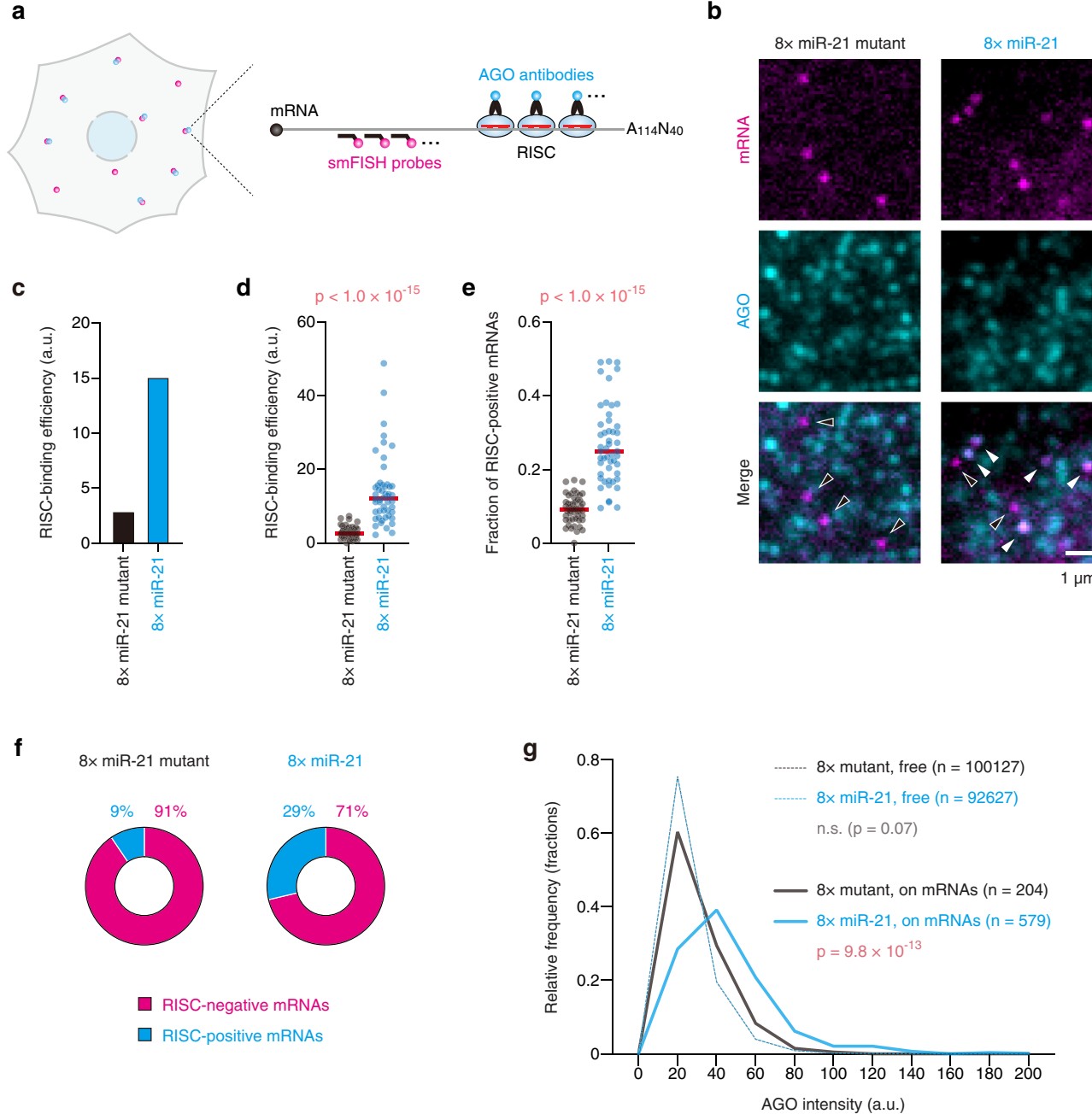

**Fig. 3 Single-molecule imaging of RISC-binding. a** Schematic of the IF-FISH experiment to visualize RISC-binding at single-mRNA resolution. Cyan and magenta spots represent RISC and reporter mRNAs, respectively. **b** The images of U2OS cells expressing the 8× miR-21 mutant reporter (left) and the 8× miR-21 reporter (right). Reporter mRNAs (magenta, top) and AGO (cyan, middle) were labeled by IF-FISH. Merged images are shown at the bottom. White and black arrowheads indicate RISC-positive and RISC-negative mRNAs, respectively. Scale bar, 1 μm. **c, d** RISC-binding mediated by miR-21. Images were analyzed using CellProfiler and FISH-quant. Then, RISC-binding efficiency was calculated by division of the total AGO intensity on mRNAs by the number of mRNAs as described in Supplementary Fig. 7 (see also Methods). The results of bulk analysis (**c**) and single-cell analysis (**d**) are shown. In **d**, each circle represents a single cell ($n = 50$ for each condition), while red lines represent the medians. The p-value of one-tailed Mann–Whitney test is shown. **e** Increase of the fraction of RISC-positive mRNAs by miR-21. The fraction of RISC-positive mRNAs was calculated as described in Supplementary Fig. 7 (see also Methods). Each circle represents a single cell ($n = 50$ for each condition), while red lines represent the medians. The p-value of one-tailed Mann–Whitney test is shown. **f** The ratio of RISC-negative and RISC-positive mRNAs. All mRNAs were classified into RISC-negative or RISC-positive mRNAs based on 3D-colocalization analysis. **g** The histogram of AGO intensity. The intensities of free AGO spots (dashed lines) and of AGO spots on mRNAs (solid lines) are shown. The p-values of Dunn's multiple comparisons test are shown. n.s., not significant. Source data are provided as a Source Data file.

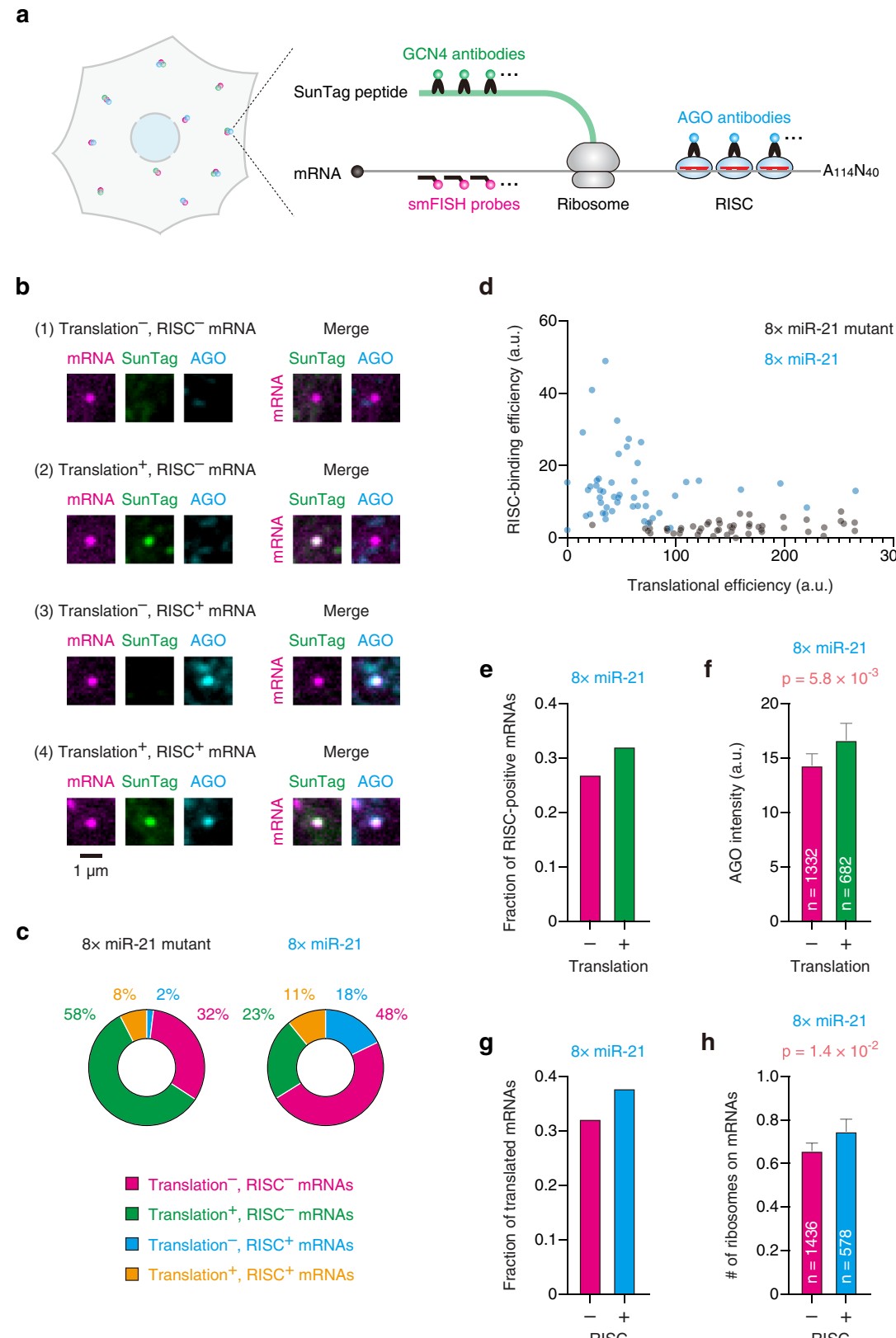

**Spatiotemporal analysis of RISC-binding, translational repression, and mRNA decay by single-mRNA imaging.** Even though RISC preferred translated mRNAs over untranslated mRNAs, if RISC repressed translation immediately, most of RISC-positive mRNAs should be untranslated mRNAs. As our data showed the opposite (Fig. 4e–h), we speculated that RISC needs a

relatively long time to repress translation. Taking advantage of our methods, which can visualize RISC-binding, translational repression, and mRNA decay, at the single-mRNA and single-cell levels, we next explored the time-course of miRNA-mediated gene silencing.

Firstly, using the methods we developed (Figs. 2a and 4a), we performed spatiotemporal analysis of RISC-binding and

**Fig. 4 Simultaneous visualization of single mRNAs, translation, and RISC-binding. a** Schematic of the SINAPS-IF-FISH experiment to visualize translation and RISC-binding simultaneously at single-mRNA resolution. Magenta, green, and cyan spots represent reporter mRNAs, SunTag peptides, and RISC, respectively. **b** The images of a RISC-negative untranslated mRNA (first row), a RISC-negative translated mRNA (second row), a RISC-positive untranslated mRNA (third row), and a RISC-positive translated mRNA (fourth row) in U2OS cells are shown. Reporter mRNAs (magenta, first column), SunTag peptides (green, second column), and AGO (cyan, third column) were labeled by SINAPS and IF-FISH. Merged images are shown on the right side. Scale bar, 1 μm. **c** The ratio of RISC-negative untranslated (magenta), RISC-negative translated (green), RISC-positive untranslated (cyan), and RISC-positive translated (orange) mRNAs. All mRNAs were classified into these four classes based on 3D-colocalization analysis. **d** A trend of negative correlation between translational efficiency and RISC-binding efficiency at the single-cell level. Images were analyzed using CellProfiler and FISH-quant. Then, translational efficiency and RISC-binding efficiency were calculated as described in Supplementary Fig. 9 (see also Methods). Each circle represents a single cell ($n = 50$ for each condition). **e, f** Translated mRNAs tend to be RISC-positive mRNAs. The fraction of RISC-positive mRNAs (**e**) and the intensity of AGO on mRNAs (**f**) are shown. In **f**, the means with SEM and the $p$-value of one-tailed Mann–Whitney test are shown. Magenta and green bars represent the values of untranslated and translated mRNAs, respectively. **g, h** RISC-positive mRNAs tend to be translated mRNAs. The fraction of translated mRNAs (**g**) and the number of ribosomes on mRNAs (**h**) are shown. In **h**, the means with SEM and the $p$-value of one-tailed Mann–Whitney test are shown. Magenta and cyan bars represent the values of RISC-negative and RISC-positive mRNAs, respectively. Source data are provided as a Source Data file.

translational repression simultaneously. In this analysis, we used the Ponasterone A (PonA)-inducible promoter, which can strictly control transcription of reporter mRNAs[44–48] (Fig. 2a). After the pulse of PonA treatment, reporter mRNAs were observed by single-mRNA imaging at three different time points (Fig. 5a). During these experiments, the outlines of the nuclei and cells, visualized by DAPI and the non-specific background signals of smFISH probes, respectively, were automatically detected by the CellProfiler algorithm[49] (Supplementary Fig. 9). Validating our experiments, the ratio of the number of cytoplasmic mRNAs to that of nuclear mRNAs increased over time after nuclear export (Fig. 5b). Notably, single-cell analysis revealed that RISC binds to cytoplasmic mRNAs immediately after the completion of PonA treatment (Fig. 5c and Supplementary Fig. 12a). These cytoplasmic mRNAs should be the mRNAs immediately after nuclear export, because most mRNAs are still in the nucleus at this time point (Fig. 5b). Single-mRNA analysis that analyzed the intensity of AGO on cytoplasmic mRNAs, confirmed that RISC-binding took place instantly (Fig. 5d and Supplementary Fig. 12c). When focusing on translational repression, however, single-cell analysis showed that RISC did not repress translation until 30 min after the PonA pulse (Fig. 5e and Supplementary Fig. 12b). This was confirmed by single-mRNA analysis, where the number of ribosomes on translated mRNAs in the cytoplasm was analyzed (Fig. 5f and Supplementary Fig. 12c). Together, these results indicated that RISC bound to mRNAs immediately after their transport from the nucleus to the cytoplasm, followed by translational repression within 30 min. The data of spatiotemporal analysis also confirmed that RISC preferred translated mRNAs over untranslated mRNAs (Fig. 5g, h).

Finally, we performed spatiotemporal analysis of mRNA decay using our reporter system (Fig. 1a). Under the control of the PonA-inducible bi-directional promoter, *Fluc* mRNAs, the internal control, and *SunTag* mRNAs, the reporter to monitor mRNA decay, were expressed for time-course experiments (Fig. 6a). As with the analysis for RISC-binding and translational repression (Fig. 5b), the ratio of the number of cytoplasmic mRNAs to that of nuclear mRNAs was increased over time (Fig. 6b), indicating that our spatiotemporal analysis worked well. Unlike translational repression, mRNA decay was not observed at 30 min after the PonA pulse (Fig. 6c). Instead, single-cell analysis showed a reduction of mRNA stability at 60 min after the PonA pulse, indicating that RISC induced mRNA decay within 60 min after the binding to mRNA. Additional time points will reveal a more precise time-course of miRNA-mediated gene silencing, e.g., the time lag between translational repression and mRNA decay, in the future.

## Discussion

Historically, miRNAs have been studied primarily by ensemble methods, where a bulk collection of molecules was measured

outside cells[14–19,23,29–32,41–43]. Although recent studies provided several valuable methods to analyze the function of miRNAs more precisely[50–55], the behavior of individual molecules inside cells, as well as their spatiotemporal regulation, remains mostly unknown. In this study, we developed a series of single-molecule methods, which enabled us to image each step of miRNA-mediated gene silencing: RISC-binding, translational repression, and mRNA decay, inside cells. Our methods, which overcame the limitation of canonical methods, provided novel insights into when, where, and how miRNAs work inside cells (Fig. 7): (1) RISC bound to mRNAs immediately after their transport from the nucleus to the cytoplasm; (2) RISC preferred translated mRNAs over untranslated mRNAs; (3) RISC repressed translation of mRNAs within 30 min after the binding; (4) RISC reduced both the number of translated mRNAs and the number of ribosomes on translated mRNAs; (5) RISC completely halted translation of mRNAs in some cells; (6) RISC induced mRNA decay within 60 min after binding to mRNAs.

We found that RISC bound to mRNAs immediately after nuclear export (Fig. 5 and Supplementary Fig. 12). Notably, the basal level of translation was not fully active at this time point (Fig. 5e and Supplementary Fig. 12b, see the data of the negative control). Together, these findings indicate that RISC bound to mRNAs when translation was not fully active, and kept them from being activated. This way of silencing would be more efficient than after translation was underway. We also found that RISC preferentially bound to translated mRNAs rather than untranslated mRNAs (Fig. 4 and Supplementary Fig. 11). Were RISC to recognize translated mRNAs and untranslated mRNAs with the same efficiency, this would be a waste of RISC, as RISC does not need to bind to untranslated mRNAs. Thus, we speculate that this preference would contribute to the economical use of RISC, the number of which is limited inside cells[35,56]. The mechanism that enables RISC to preferentially bind to translated mRNAs should be investigated in the future. It is noteworthy that small interfering RNAs (siRNAs) also prefer translated mRNAs[57]; a recent study revealed that translating ribosomes unfold mRNA structures and unmask siRNA sites, thereby promoting interactions between target mRNAs and siRNAs. Although most miRNA sites are located within the 3′ UTR, where ribosomes do not access, as base-pairing within RNAs can occur over large distances[58], miRNAs might prefer translated mRNAs through a similar mechanism.

RISC generally targets the 3′ UTR of mRNAs. It has been proposed that this is because RISC targeting regions other than the 3′ UTR may be removed from mRNAs by translating ribosomes before it represses translation[13,59,60]. Notably, this model is based on the assumption that RISC may need a longer time to repress translation than the speed of translation. However,

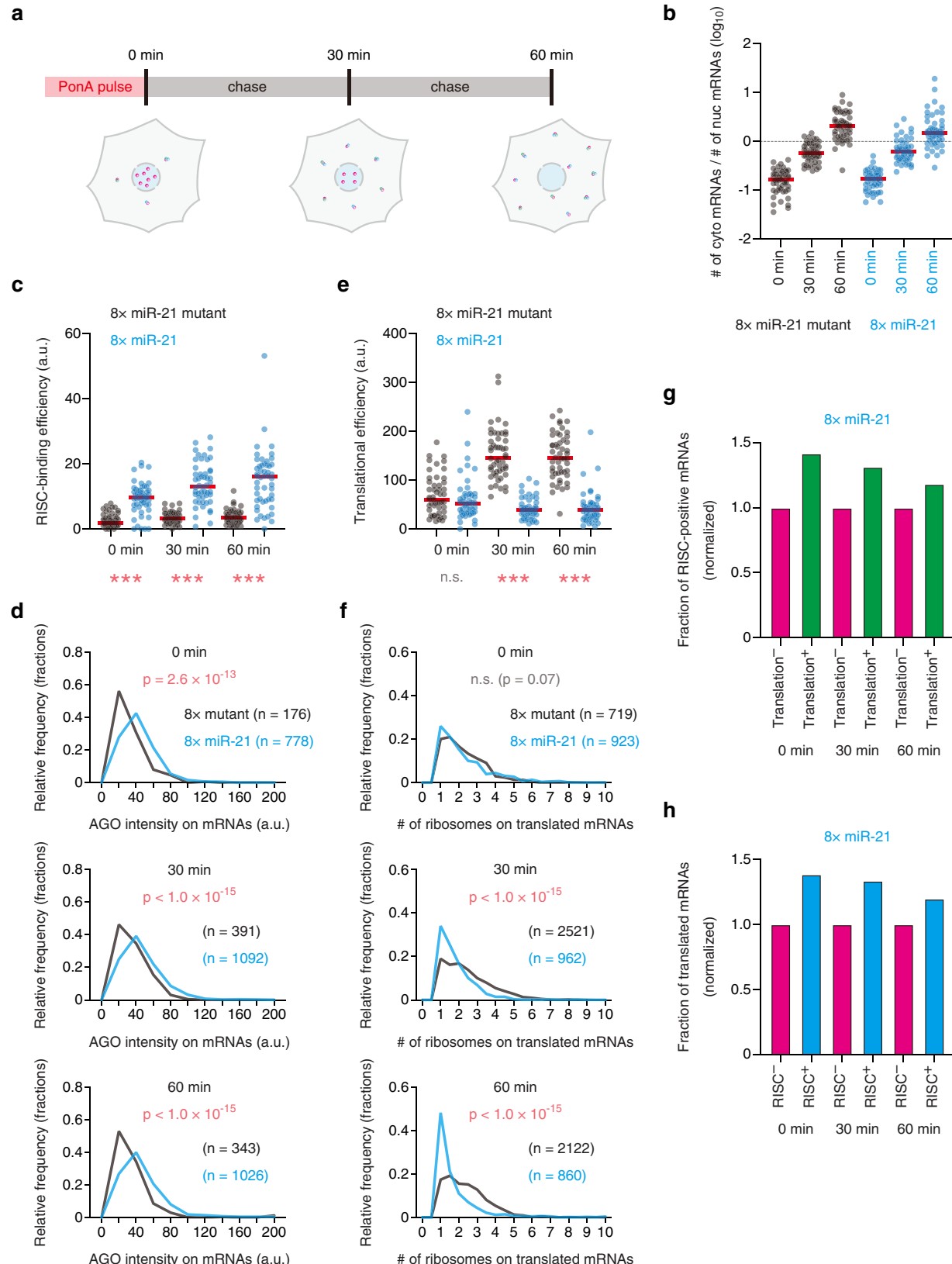

the time span from RISC-binding to translational repression has been unknown. In this study, our spatiotemporal analysis revealed that RISC needs ~30 min to repress translation after the binding to mRNAs (Fig. 5). As this is much slower than translation initiation rates, typically faster than ~1 per min[61], our finding provides a missing piece to explain why RISC generally uses the 3′ UTR of mRNAs: translational repression by RISC is too slow to compete with ribosomes. Nonetheless, translation initiation rates differ depending on each mRNA species[61] and are dynamically regulated in response to cellular contexts[62–64].

**Fig. 5 Spatiotemporal analysis of RISC-binding and translational repression by single-mRNA imaging. a** Schematic of spatiotemporal analysis of RISC-binding and translational repression by single-mRNA imaging. In pulse-chase experiments, IF-FISH and SINAPS were performed to visualize RISC-binding and translation at single-mRNA resolution. Magenta, green, and cyan spots represent reporter mRNAs, SunTag peptides, and RISC, respectively. **b** Transport of reporter mRNAs from the nucleus to the cytoplasm. In pulse-chase experiments, reporter mRNAs were labeled by smFISH. The ratios of the number of cytoplasmic mRNAs to that of nuclear mRNAs are shown. Each circle represents a single cell ($n = 50$ for each condition), while red lines represent the medians. cyto, cytoplasmic. nuc, nuclear. **c**–**f** Time-course analysis of RISC-binding (**c** and **d**) and translational repression (**e** and **f**) by single-mRNA imaging. Images were analyzed using CellProfiler and FISH-quant. Then, RISC-binding efficiency (**c**), the intensity of AGO on mRNAs (**d**), translational efficiency (**e**), and the number of ribosomes on translated mRNAs (**f**) were calculated as described in Supplementary Fig. 9 (see also Methods). In **c** and **e**, each circle represents a single cell ($n = 50$ for each condition), while red lines represent the medians. The $p$-values of Dunn's multiple comparisons test are shown. *** and n.s. represent $p < 0.001$ and not significant ($p > 0.05$), respectively. $p$-values in **c** were $3.0 \times 10^{-10}$ (0 min), $1.6 \times 10^{-13}$ (30 min), and $6.1 \times 10^{-13}$ (60 min), while those in **e** were $> 0.99$ (0 min), $<1.0 \times 10^{-15}$ (30 min), and $<1.0 \times 10^{-15}$ (60 min). **g** Translated mRNAs tend to be RISC-positive mRNAs at all time points. The fraction of RISC-positive mRNAs is shown. Magenta and green bars represent the values of untranslated and translated mRNAs, respectively. **h** RISC-positive mRNAs tend to be translated mRNAs at all time points. The fraction of translated mRNAs is shown. Magenta and cyan bars represent the values of RISC-negative and RISC-positive mRNAs, respectively. Source data are provided as a Source Data file.

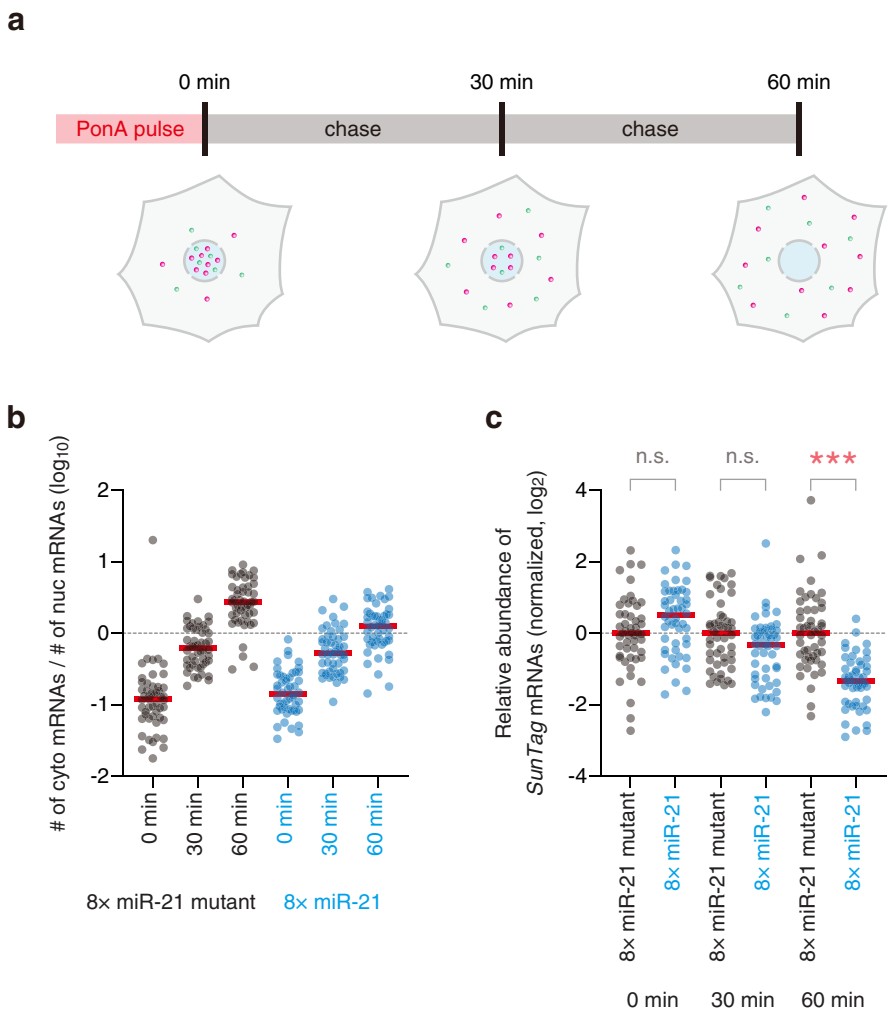

**Fig. 6 Spatiotemporal analysis of mRNA decay by single-mRNA imaging. a** Schematic of spatiotemporal analysis of mRNA decay by single-mRNA imaging. In pulse-chase experiments, smFISH was performed to visualize mRNA decay at single-mRNA resolution. Green and magenta spots represent *SunTag* and *Fluc* mRNAs, respectively. **b** Transport of reporter mRNAs from the nucleus to the cytoplasm. In pulse-chase experiments, reporter mRNAs were labeled by smFISH. The ratios of the number of cytoplasmic mRNAs to that of nuclear mRNAs are shown. Each circle represents a single cell ($n = 50$ for each condition), while red lines represent the medians. cyto, cytoplasmic. nuc, nuclear. **c** Time-course analysis of mRNA decay by single-mRNA imaging. Images were analyzed using CellProfiler and FISH-quant. Then, mRNA stability (the relative abundance of *SunTag* mRNAs to *Fluc* mRNAs) was calculated as described in Supplementary Fig. 2 (see also Methods), followed by normalization to the value of the negative control at each time point. Each circle represents a single cell ($n = 50$ for each condition), while red lines represent the medians. The results of Dunn's multiple comparisons test are shown. *** and n.s. represent $p < 0.001$ and not significant ($p > 0.05$), respectively. $p$-values were 0.24 (0 min), 0.11 (30 min), and $6.4 \times 10^{-10}$ (60 min). Note that time-course data of RISC-binding, translational repression, and mRNA decay were analyzed using exactly the same statistical test (Dunn's multiple comparisons test). Source data are provided as a Source Data file.

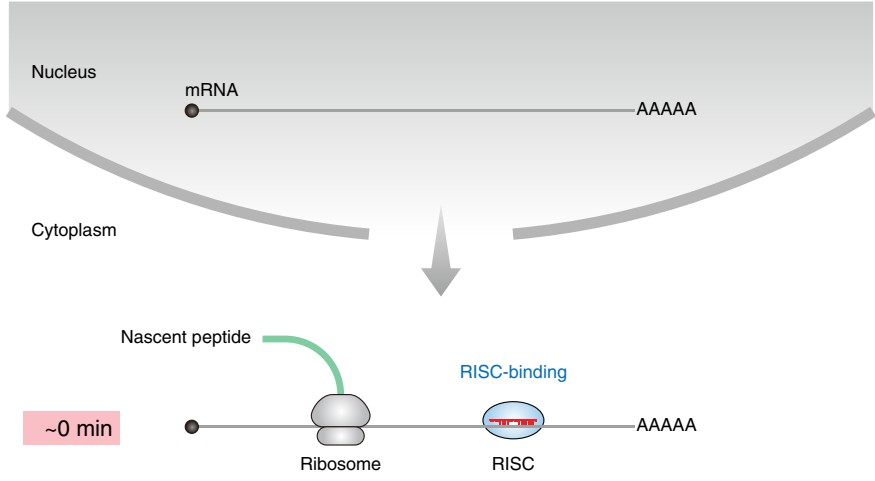

Nucleus

mRNA ●━━━━━━━━━━━━━━━━━ AAAAA

Cytoplasm

Nascent peptide

RISC-binding

~0 min    ●━━━━━━━ Ribosome ━━━━ RISC ━━━ AAAAA

1) RISC binds to target mRNAs immediately after nuclear export

2) RISC prefers translated mRNAs over untranslated mRNAs

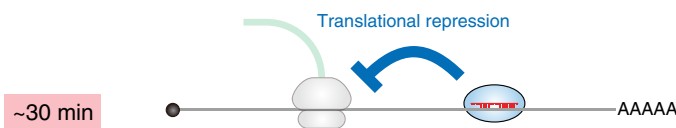

Translational repression

~30 min

3) After the binding to mRNAs, RISC represses their translation within 30 min

4) RISC reduces the # of translated mRNAs and the # of ribosomes on translated mRNAs

5) In some cells, RISC can completely halt translation of target mRNAs

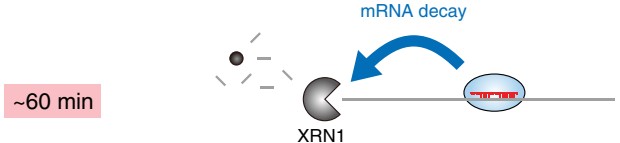

mRNA decay

~60 min    XRN1

6) After the binding to mRNAs, RISC induces their decay within 60 min

**Fig. 7 A model of miRNA-mediated gene silencing; findings from single-molecule imaging inside cells.** After mRNAs are exported from the nucleus to the cytoplasm, RISC binds to them immediately. RISC preferentially binds to translated mRNAs rather than untranslated mRNAs. Then, RISC represses translation within 30 min after the binding to mRNAs. This action of RISC reduces the number of translated mRNAs inside cells, as well as the number of ribosomes on translated mRNAs. In some cells, RISC can completely halt translation of target mRNAs. Subsequently, RISC induces mRNA decay within 60 min after the binding to mRNAs.

Therefore, our model does not exclude noncanonical miRNA sites within the 5′ UTR and the ORF. The reason why RISC needs ~30 min to repress translation should be addressed in future work.

mRNA regulation is a fundamental step in gene regulation, thereby influencing a wide variety of biological processes and diseases. To regulate mRNAs, various RNA-binding proteins (RBPs) associate with mRNAs, which trigger the stabilization or degradation of mRNAs, the activation or repression of translation, or the translocation of mRNAs to specific areas. The methods to analyze such mRNA regulation in situ, however, have been limited. Since our methods, which make it possible to visualize single mRNAs, translation, and RISC-binding simultaneously with single-molecule resolution inside cells, can potentially be customized for other RBPs, this methodology could be a valuable framework for studying mRNA regulation in the future.

## Methods

### Plasmid construction

*The reporter plasmid for miRNA-mediated mRNA decay.* The plasmid pPonA-BI-Gl-NORM-LacZA TER-LacZB (Addgene plasmid # 86212), which expresses two different mRNAs (*Gl-NORM-LacZA* and *Gl-TER-LacZB*) under the control of the PonA-inducible bi-directional promoter[46], was used as the backbone sequence. The Gl-TER-LacZB sequence was replaced by the Fluc sequenc[65,66], while Gl-NORM-LacZA sequence was replaced by the SunTag sequence followed by the AID degron[28]. The degron sequence was inserted in order to visualize translational repression, but not to visualize of mRNA decay. Eight miR-21 sites (or eight miR-21 mutant sites) were inserted into the 3′ UTR of the *SunTag* mRNA to recapitulate mRNA decay by miRNAs (Fig. 1a). The miR-21 sites were designed so that they form base-pairs with both the seed region and the 3′ supplemental region of miR-21[11–13] (Supplementary Fig. 1d, left). Although the seed region is primarily responsible for miRNA targeting, a recent study revealed that RISC can bind to mRNAs only by base-pairing of the 3′ region of miRNAs without involvement of the seed region[67]. As such, the miR-21 mutant sites were designed so that neither the seed region nor the 3′ supplemental region forms base-pairs with them (Supplementary Fig. 1d, right). Although the backbone sequence originally had the SV40 poly(A) signals, they were replaced by the bGH poly(A) signals.

*The reporter plasmid for miRNA-mediated translational repression*. The plasmid for miRNA-mediated mRNA decay was used as the backbone sequence. The bGH poly(A) signal for the *SunTag* mRNA was replaced by the $A_{114}$-$N_{40}$-HhR sequence, which protects mRNAs from decay; the $A_{114}$-$N_{40}$ sequence consists of a 114-nt poly(A) sequence (A114) followed by a 40-nt unrelated sequence (N40), which protects mRNAs against deadenylation, the first step of miRNA-mediated mRNA decay[29–32]. To eliminate the unwanted RISC-binding to reporter mRNAs independent of miR-21, all potential miRNA sites (8mer, 7mer, and 6mer) of top 30 most abundant miRNAs (Supplementary Fig. 1b, c) were removed from the 3′ UTR of the reporter mRNAs.

*The reporter plasmid for RISC-binding*. To investigate the sensitivity of our method to RISC, the 8× miR-21 sequence within the plasmid for miRNA-mediated translational repression was replaced by a single miR-21 site or its mutant site. Reporter plasmids were purified using NucleoBond Xtra Midi kit (MACHEREY-NAGEL, 740410) for the purpose of nucleofection.

**Cell culture**. The human U2OS cells stably expressing VgEcR and RXR, which enable PonA-inducible transcription[48], were cultured at 37 °C and 5% $CO_2$ in DMEM (Corning, 10-013-CV) supplemented with 10% FBS (R&D Systems, S11150H) and 1% Antibiotic-Antimycotic (Thermo Fisher, 15240-062).

**Nucleofection**. U2OS cells were briefly rinsed with DPBS (Corning, 21-031-CV), followed by the treatment with Trypsin-EDTA (Thermo Fisher, 25300-054) to detach them. After adding culture media to neutralize trypsinization, cells were centrifuged at $300 \times g$ for 1 min. The pellets (approx. $1 \times 10^6$ cells) were resuspended with 100 μl of Ingenio Electroporation Solution (Mirus, MIR50115) containing 2 μg of reporter plasmids. Then, nucleofection was performed in the electroporation cuvette (Mirus, MIR50115) using Nucleofector II (Lonza). Nucleofected cells were cultured on the coverslips (Thermo Fisher, 12-545-81) coated with collagen (Cell Applications, 125-50) in culture media.

**Drug treatment**. One day after nucleofection, U2OS cells were treated with 20 μM PonA (Santa Cruz, sc-202768A) for 30 min to induce transcription of reporter mRNAs. After washing cells with culture media, cells were incubated for 1 h. Subsequently, cells were fixed with 4% paraformaldehyde (Electron Microscopy Sciences, 15714) in 1× PBS (MilliporeSigma, 11666789001) for 10 min. For puromycin experiments (Supplementary Fig. 5), cells were treated with 100 μg/ml puromycin (MilliporeSigma, 540222-25MG) from the beginning of PonA treatment until fixation. For cycloheximide experiments (Supplementary Fig. 11), cells were treated with 100 μg/ml cycloheximide (MilliporeSigma, C7698-5G) for 30 min before fixation.

**Pulse-chase experiment**. One day after nucleofection, U2OS cells were treated with 20 μM PonA for 30 min to induce transcription of reporter mRNAs. After washing cells with culture media, cells were fixed at 0, 30, and 60 min after PonA treatment with 4% paraformaldehyde in 1× PBS for 10 min.

**Single-molecule fluorescence in situ hybridization (smFISH)**. Fixed cells were permeabilized with 0.1% Triton X-100 (MilliporeSigma, T9284) in 1× PBS for 10 min, followed by washing with 1× PBS. Subsequently, cells were incubated with the pre-hybridization solution containing 10% deionized formamide (Thermo Fisher, AC205821000), 2× SSC (MilliporeSigma, 11666681001), 0.5% UltraPure BSA (Thermo Fisher, AM2618), and 40 U/ml SUPERase In RNase Inhibitor (Thermo Fisher, AM2696) for 30 min. Then, cells were incubated with the hybridization solution containing 10% deionized formamide, 2× SSC, 10% dextran sulfate (MilliporeSigma, D8906), 1 mg/ml competitor tRNA (MilliporeSigma, 10109541001), 0.05% UltraPure BSA, 40 U/ml SUPERase In RNase Inhibitor, and 50 nM smFISH probes for 3 h at 37 °C. After washing with 10% deionized formamide in 2× SSC, followed by washing with 2× SSC, coverslips were mounted onto glass slides (Thermo Fisher, 3051-002) using ProLong Diamond Antifade Mountant with DAPI (Thermo Fisher, P36962). The smFISH probes toward *Fluc* mRNAs were designed using Stellaris Probe Designer version 4.2 (Biosearch Technologies). The smFISH probes conjugated with Quasar 570 toward *Fluc* mRNAs and the smFISH probes conjugated with Quasar 670 toward *SunTag* mRNAs[22] were synthesized by Biosearch Technologies. The sequences of smFISH probes are listed in the Supplementary Table 1.

**Single-molecule imaging of nascent peptides (SINAPS)**. Fixed cells were permeabilized and pre-hybridized as described in the smFISH section. Then, cells were incubated with the hybridization solution containing 10% deionized formamide, 2× SSC, 10% dextran sulfate, 1 mg/ml competitor tRNA, 0.05% UltraPure BSA, 40 U/ml SUPERase In RNase Inhibitor, 50 nM smFISH probes conjugated with Quasar 570 toward *SunTag* mRNAs[22], and 10 μg/ml anti-GCN4 Rabbit antibody (Absolute Antibody, AB00436-23.0) for 3 h at 37 °C. Subsequently, cells were washed with 10% deionized formamide in 2× SSC, followed by incubation with 10% deionized formamide in 2× SSC supplemented with 2 μg/ml Goat anti-Rabbit IgG conjugated with Alexa Fluor 488 (Thermo Fisher, A-11034) for 30 min at 37 °C. After washing with 2×

SSC, coverslips were mounted onto glass slides using ProLong Diamond Antifade Mountant with DAPI.

**IF-FISH (fluorescence in situ hybridization)**. Fixed cells were permeabilized with 0.1% Triton X-100 in 1× PBS, followed by washing with 1× PBS. Subsequently, cells were incubated with the blocking buffer (1× PBS, 0.02% Triton X-100, 0.5% UltraPure BSA, and 40 U/ml SUPERase In RNase Inhibitor) for 30 min. Then, cells were incubated with the blocking buffer supplemented with 24 μg/ml anti-AGO2 Mouse antibody (FUJIFILM Wako Pure Chemical, 015-22031, clone 4G8) for 1 h. In Supplementary Fig. 8b, anti-AGO2 Rat antibody (MilliporeSigma, MABE253, clone 11A9) was also tested at a 1:50 dilution. Cells were washed with the blocking buffer, followed by incubation with the blocking buffer supplemented with 2 μg/ml Goat anti-Mouse IgG conjugated with Alexa Fluor 647 (Thermo Fisher, A-21236) for 30 min. After washing with 1× PBS, immunostained cells were pre-hybridized and hybridized to perform smFISH as described in the smFISH section. For the experiments to visualize single mRNAs, translation, and RISC-binding simultaneously (Figs. 4 and 5), immunostained cells were pre-hybridized and hybridized to perform SINAPS as described in the SINAPS section.

**Image acquisition**. Slides were imaged on the BX63 automated wide-field fluorescence microscope (Olympus) equipped with the SOLA FISH light engines (Lumencor), the ORCA-R2 cooled digital CCD camera (Hamamatsu Photonics), the 60 × 1.35 NA super apochromat objective (Olympus, UPLSAPO60XO), and zero pixel shift filter sets: DAPI-5060C-Zero, FITC-5050A-Zero, Cy3-4040C-Zero, and Cy5-4040C-Zero (Semrock). To acquire multi-color 3D images, the microscope was controlled with MetaMorph software (Molecular Devices), where the Multi Dimensional Acquisition mode was selected. Exposure times for each color were 100–200 ms for CY5 (gain: 2), 100–200 ms for CY3 (gain: 2), 100–200 ms for FITC (gain: 2), and 5–10 ms for DAPI (gain: 0). For each color, Z-stacks spanning the entire volume of cells were acquired by imaging every 200 nm along the z-axis. Image pixel size: XY, 107.5 nm; Z, 200 nm. $n = 50$ cells for each experiment.

**Image analysis**. Images were analyzed using FISH-quant[26], an algorithm implemented in MATLAB. Briefly, after background subtraction, FISH-quant automatically detects fluorescent spots and localizes them in 3D at sub-pixel resolution by fitting 3D Gaussians. This provides the number of spots inside cells, the intensity of each spot, and the 3D position of each spot[26]. To distinguish nuclear and cytoplasmic areas, the nuclei and cells were visualized by DAPI and the non-specific background signals of smFISH probes, respectively. Their outlines were automatically detected by the CellProfiler algorithm[49], followed by conversion to the outline files compatible with FISH-quant. Based on these outlines, all spots were classified into "nuclear" and "cytoplasmic". To prepare images for figures (Figs. 1c, 2c, 3b, 4b and Supplementary Fig. 8b), raw images were processed using ImageJ software (Version: 2.1.0/1.53c).

**Colocalization analysis**. Colocalization in 3D was analyzed using FISH-quant[26]. First, the average drift between different colors was calculated and corrected. Then, using the 3D positions of spots, their 3D distances were calculated. When two spots in different colors were localized within the maximum allowed distance (mRNA-SunTag, 500 nm; mRNA-AGO, 250 nm), these spots were considered colocalized. Compared with SunTag IF spots (Supplementary Fig. 6b, median, ~250 spots per cell), AGO has many more IF spots (Supplementary Fig. 8c, median, ~2000 spots per cell), which cause the non-specific coincidence of the 3D positions of mRNAs and AGO by chance. To minimize such false colocalizations, we adopted the shorter maximum allowed distance for mRNA-AGO colocalization. Based on the 3D-colocalization with SunTag and AGO, mRNAs were classified into "translated", "untranslated", "RISC-positive", and "RISC-negative". Likewise, SunTag and AGO spots were also classified into "on mRNAs" and "free", depending on the colocalization with mRNAs.

**Data analysis**

*Profiling miRNAs expressed in U2OS cells*. The small RNA-seq data of U2OS cells (GEO accession: GSM416754)[23] was reanalyzed as described previously[65].

*Profiling AGO proteins expressed in U2OS cells*. From the proteome data of U2OS cells[35], the copies per cell values of AGO1 (UniProtKB: Q9UL18), AGO2 (UniProtKB: Q9UKV8), AGO3 (UniProtKB: Q9H9G7), and AGO4 (UniProtKB: Q9HCK5) were extracted.

*Data analysis for miRNA-mediated mRNA decay*. The mRNA stability (the relative abundance of *SunTag* mRNAs to *Fluc* mRNAs) of each cell, which is used for single-cell analysis (Figs. 1e and 6c), was calculated by Eq. 1, where $S(k)$ is the mRNA stability of the $k^{th}$ cell. $M_{\text{Fluc,cyto}}(k)$ and $M_{\text{Sun,cyto}}(k)$ are the number of *Fluc* mRNAs and *SunTag* mRNAs, respectively, in the cytoplasm of the $k^{th}$ cell.

$$S(k) = \frac{M_{\text{Sun,cyto}}(k)}{M_{\text{Fluc,cyto}}(k)} \qquad (1)$$

The mRNA stability of the cell population (50 cells), $S_{bulk}$, which is used for bulk analysis (Fig. 1d), was calculated by Eq. 2.

$$S_{bulk} = \frac{\sum_{k=1}^{50} M_{Sun,cyto}(k)}{\sum_{k=1}^{50} M_{Fluc,cyto}(k)} \quad (2)$$

*Data analysis for miRNA-mediated translational repression.* The translational efficiency of each cell, which is used for single-cell analysis (Figs. 2e, 5e, and Supplementary Fig. 5b, e), was calculated by Eq. 3, where $T_{eff}(k)$ is the translational efficiency of the $k^{th}$ cell. $I_{Sun,coloc,cyto}(k)$ is the total intensity of SunTag spots on mRNAs in the cytoplasm of the $k^{th}$ cell.

$$T_{eff}(k) = \frac{I_{Sun,coloc,cyto}(k)}{M_{Sun,cyto}(k)} \quad (3)$$

The translational efficiency of the cell population (50 cells), $T_{eff,bulk}$, which is used for bulk analysis (Fig. 2d and Supplementary Fig. 5a, d), was calculated by Eq. 4.

$$T_{eff,bulk} = \frac{\sum_{k=1}^{50} I_{Sun,coloc,cyto}(k)}{\sum_{k=1}^{50} M_{Sun,cyto}(k)} \quad (4)$$

The fraction of translated mRNAs of each cell, which is used for single-cell analysis (Fig. 2f, Supplementary Fig. 5c, f and 12b), was calculated by Eq. 5, where $T_{fra}(k)$ is the fraction of translated mRNAs of the $k^{th}$ cell. $M_{Sun,coloc,cyto}(k)$ is the number of translated *SunTag* mRNAs in the cytoplasm of the $k^{th}$ cell.

$$T_{fra}(k) = \frac{M_{Sun,coloc,cyto}(k)}{M_{Sun,cyto}(k)} \quad (5)$$

The number of ribosomes on translated mRNAs, $R$, which is used for single-molecule analysis (Fig. 5f and Supplementary Fig. 6e), was calculated by Eq. 6. $i_{MED,Sun,free,cyto}$ is the median of the intensities of free SunTag spots in the cytoplasm, while $i_{Sun,coloc,cyto}$ is the intensity of each SunTag spot on mRNAs in the cytoplasm. Bright SunTag spots on mRNAs should contain partial SunTag peptides, which had not been fully translated, as well as full-length SunTag peptides. Thus, the exact number of ribosomes on translated mRNAs should be larger than our values[22].

$$R = \frac{i_{Sun,coloc,cyto}}{i_{MED,Sun,free,cyto}} \quad (6)$$

*Data analysis for RISC-binding.* The RISC-binding efficiency of each cell, which is used for single-cell analysis (Figs. 3d and 5c; Supplementary Figs. 8h and 11d), was calculated by Eq. 7, where $A_{eff}(k)$ is the RISC-binding efficiency of the $k^{th}$ cell. $I_{AGO,coloc,cyto}(k)$ is the total intensity of AGO spots on mRNAs in the cytoplasm of the $k^{th}$ cell.

$$A_{eff}(k) = \frac{I_{AGO,coloc,cyto}(k)}{M_{Sun,cyto}(k)} \quad (7)$$

The RISC-binding efficiency of the cell population (50 cells), $A_{eff,bulk}$, which is used for bulk analysis (Fig. 3c ; Supplementary Figs. 8g and 11c), was calculated by Eq. 8.

$$A_{eff,bulk} = \frac{\sum_{k=1}^{50} I_{AGO,coloc,cyto}(k)}{\sum_{k=1}^{50} M_{Sun,cyto}(k)} \quad (8)$$

The fraction of RISC-positive mRNAs of each cell, which is used for single-cell analysis (Fig. 3e and Supplementary Fig. 12a), was calculated by Eq. 9, where $A_{fra}(k)$ is the fraction of RISC-positive mRNAs of the $k^{th}$ cell. $M_{AGO,coloc,cyto}(k)$ is the number of RISC-positive *SunTag* mRNAs in the cytoplasm of the $k^{th}$ cell.

$$A_{fra}(k) = \frac{M_{AGO,coloc,cyto}(k)}{M_{Sun,cyto}(k)} \quad (9)$$

*Data analysis for mRNA export.* The mRNA export efficiency of each cell, which is used for single-cell analysis (Figs. 5b and 6b), was calculated by Eq. 10, where $E(k)$ is the mRNA export efficiency of the $k^{th}$ cell. $M_{Sun,nuc}(k)$ and $M_{Sun,cyto}(k)$ are the number of *SunTag* mRNAs in the nucleus and the cytoplasm, respectively, of the $k^{th}$ cell.

$$E(k) = \frac{M_{Sun,cyto}(k)}{M_{Sun,nuc}(k)} \quad (10)$$

**Statistical analysis**. One-tailed Mann–Whitney tests were performed in Figs. 1e, 2e, 2f, 3d, 3e, 4f, 4h, Supplementary Figs. 3a, 3b, 5b, 5c, 5e, 5f, 6a, 6b, 6e, 10a, 10b, 11d, and 11e, while Dunn's multiple comparisons tests were performed in Figs. 2h, 3g, 5c, 5d, 5e, 5f, 6c, Supplementary Figs. 6c, 6d, 8c, 8d, 8e, 8f, 8h, 12a, and 12b. One-tailed paired *t*-tests were performed in Supplementary Fig. 11a, b. ***, *, and n.s. represent $p < 0.001$, $p < 0.05$, and not significant ($p > 0.05$), respectively. These statistical tests, calculation of Pearson correlation coefficient ($r$) (Supplementary Fig. 3c), and simple linear regression (Supplementary Fig. 3c) were performed using GraphPad Prism (Version: 8 and 9), which is also used to create all graphs in this study.

**Reporting summary**. Further information on research design is available in the Nature Research Reporting Summary linked to this article.

## Data availability
The small RNA-seq data analyzed in this study are available in the GEO database under accession code GSM416754[23]. The proteome data used in this study are available at Beck et al., 2011 [https://www.embopress.org/doi/full/10.1038/msb.2011.82][35]. The imaging data generated in this study are available from the corresponding authors on reasonable request. Source data are provided with this paper.

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

## Acknowledgements

We are grateful to Yukihide Tomari for providing pAWH-Rluc-let-7-A₁₁₄-N₄₀-HhR. We thank Xiuhua Meng and Melissa Lopez-Jones for technical assistance, Hanae Sato for sharing the protocols of smFISH and SINAPS, and Carolina Eliscovich for sharing the protocol for IF-FISH. We are grateful to Keisuke Shoji for supporting the analysis of small RNA-seq data. We thank David P. Bartel and all the members of the Robert H. Singer laboratory for insightful discussion and critical comments. The authors thank Timothy J. Stasevich for sharing preliminary data. This work was supported by NIH grants R01NS083085 and R35GM136296 (to R.H.S.), JSPS Overseas Research Fellow-ships (to H.K.), JBS Osamu Hayaishi Memorial Scholarship for Study Abroad (to H.K.), and JST PRESTO Grant JPMJPR20E7 (to H.K.).

## Author contributions

H.K. and R.H.S. designed the project. H.K. performed experiments and analyzed data. H.K. and R.H.S. wrote the manuscript.

## Competing interests

The authors declare no competing interests.
