## [Peer Review File · Nature Communications]

Point-by-point response to the Reviewers' comments

Reviewer #1:

The manuscript by Kobayashi et al. presents a study on the single molecule imaging of microRNA-mediated gene silencing in the cell. The authors use smFISH, labeling of nascent peptide with SunTag and antibody-labelled hAGO2 protein to follow mRNA decay and translational repression of miR21-RISC targeted or control mRNAs in fixed U2OS cells. The manuscript is for the most part written clearly, uses variation of previously described methods and approaches to test relevant biological question. The figures and data presented are is easy to follow and majority of the authors conclusions are substantiated with the experiments. That said, the majority of presented data are confirming previous studies from multiple model systems and multiple groups (Sonenberg, Filipowicz, Green, Giraldez, and Izaurralde among the others) arguing that translational repression precedes mRNA decay. Manuscript in this respect lacks clear novelty in either methodological or mechanistical approach.

We appreciate the Reviewer's positive comments and valuable feedback. As the Reviewer mentioned, we presented some single-cell and single-molecule data that confirm some previous studies. The purpose of these data, such as translational repression followed by mRNA decay, is to demonstrate that our methods work as expected. However these data may have obscured the novelty of our work. To address this concern, we have carefully modified our manuscript to emphasize our novel findings (please see the Discussion section). Moreover, according to the Reviewer's comments, we have now added experiments, analyses, and discussions that reinforce the novelty of our work (please see below).

The incremental novelty is arguing for the preferential binding of miRISCs to translated mRNAs rather than untranslated mRNAs (Fig 4). However this claim is problematic as it is not obvious from the figure 4c or S10 that this is the case. There is no control experiment to support this claim(ie. using non-coding target mRNA or inhibitors).

We appreciate the Reviewer for pointing this out. Please note that the preference of RISC toward translated mRNAs is not dramatic. That's why this is not obvious from the Fig. 4c and Fig. S12c. However, when we analyzed translated and untranslated mRNAs separately at the single-mRNA level (a unique contribution of this approach), these analyses revealed the preference of RISC toward translated mRNAs (Fig. 4e-h; Fig. 5g-h). To address the Reviewer's concern, we have now confirmed that this preference is statistically significant (a and b, below). Moreover, as suggested by the Reviewer's comment, we have also added

experiments with a translation inhibitor. These experiments confirmed that RISC signals on mRNAs were attenuated by the treatment with cycloheximide (c and d, below), supporting that RISC prefers translated mRNAs. We have now provided these additional data in the

revised manuscript (pages 10-11; Fig. S11a-d).

Additionally, there is a substantial amount of targeted mRNAs that are not bound by miRISC and that are not translated in mentioned figures. This number is almost constant in fig S10 ~55% for targeted mRNA in comparison to non-targeted mRNA - 64% at time 0 and 33% at 60%. It almost seems that numbers in targeted mRNAs do not change over time frame of the experiment (0-60min). This is never discussed and it is possible, based on the previous knowledge of miRNA-mediated gene silencing, that non-translated and non-miRISC-bound targeted mRNAs are already silenced, released from miRISC and being decayed over the course of time.

We apologize for an insufficient discussion about these data. First, as the Reviewer pointed out, the number of RISC-positive mRNAs and RISC-negative mRNAs were constant over time for targeted mRNAs (Fig. S12c, right). We speculate this is because RISC-binding reaches a plateau as early as our first point. We performed additional experiments to investigate the detection threshold of RISC signals by our method – we made new reporter mRNAs that have a single miR-21 site or its mutant site and compared RISC signals on each reporter. Although there was a small but detectable difference between them, these analyses revealed that our method is not robust enough to detect single RISC molecules unequivocally (e and f, below). Therefore, in addition to ~30% of mRNAs that are RISC-positive (Fig. S12c,

right), more mRNAs would likely be bound by RISC. This would explain in part why there is a substantial number of targeted mRNAs that are RISC (-) translation (-). In addition, we fully agree with the Reviewer – it is possible that a part of RISC (-) translation (-) mRNAs is already silenced and released from RISC. We have now added this discussion in the revised manuscript (the Results section, pages 9 and 10). The results of additional experiments are in Fig. S8g and S8h.

Regarding translation, the number of translated mRNAs was ~30% at 0 min and ~60% at 30 and 60 min for non-targeted mRNAs (Fig. S12c, left). These data indicate that translation is not fully active immediately after nuclear transport. On the other hand, as the Reviewer pointed out, this number was almost constant over time for targeted mRNAs, ~30% (Fig. S12c, right). This would be because RISC binds to mRNAs immediately after nuclear export, when translation is not fully active, and keeps them from being fully activated. This way of silencing would be more efficient than after translation is fully activated. We have also mentioned these points in the Discussion (pages 14-15).

In this respect one would like to consider additional experiments that would potentially present novelty in the mechanistic model of miRISC-mediated translational repression.

As described above, we have now performed additional experiments and analyses, which reinforce the novelty of our work. In this study, we developed novel methods to image miRNA-mediated gene silencing in situ inside cells and provided novel findings, such as 1) RISC bound to mRNAs immediately after their nuclear export; 2) RISC preferred translated mRNAs over untranslated mRNAs; 3) RISC repressed translation of mRNAs within 30 min after the binding; 4) RISC reduced both the number of translated mRNAs and the number of ribosomes on translated mRNAs. In addition, we would like to note that this is the first study that succeeded in imaging miRNA-mediated gene silencing at the single-molecule level, and hence provided quantitative data on molecular populations in cells. We have now added a more careful discussion about the results of the single-molecule analyses. We believe that our revised manuscript has been significantly improved as a result of these critiques.

Reviewer #2:

In this study by Kobayashi and Singer, the authors use single molecule imaging tools to study the mechanism of gene silencing by Argonaute in complex with a miRNA. While decades of work has unraveled many aspects of miRNA function, several key questions remained unanswered due to technical limitations of bulk analysis methods. Here, the authors use cutting edge imaging techniques to address several of these key questions. Overall, the paper is important for the field, clearly written and the conclusions are supported by the data.

The impact of the paper is probably more due to the new biological insights, than due to method development per se, as the method could be considered an application of a previously developed method (e.g. Wu et al., 2016, Science) to a new biological problem. Nonetheless, adaptation of the earlier method to study miRNAs still constitutes an advance, and the authors add visualization of an RBP (AGO) binding to single mRNAs as an additional technical advance that is especially exciting.

We are most grateful to the Reviewer for positive comments on our work.

I do have several important concerns that should be addressed before publication.

“we added the anti-decay sequence, A114-N40 (28, 29), to the end of the 3' UTR” Can the authors state which type of mRNA decay this sequence protects against and show whether addition of this sequence does in fact inhibit decay as suggested? Also, I wonder if preventing the decay of the reporter mRNA using this sequence could affect translation? It would be nice to confirm the results in the absence of this sequence.

We appreciate the Reviewer's helpful suggestion. The anti-decay sequence A114-N40 consists of a 114-nt poly(A) sequence (A114) containing a 40-nt unrelated sequence (N40),

which protects mRNAs against deadenylation, the first step of miRNA-mediated mRNA decay (PMID: 22117217; PMID: 23123195; PMID: 25280104; PMID: 25280105). Consistent with previous studies, we readily found miR-21 reporter-expressing cells where expression levels were comparable to control cells (Fig. S6a). In the revised manuscript, we have now described the A114-N40 sequence in more detail (the Results section, page 5; the Methods section, page 18).

In response to the second question, we performed additional experiments and confirmed that translation looks similar either in the presence or absence of the A114-N40 sequence under our experimental conditions (a, b, and c, below). We have now added these results in the revised manuscript (the Results section, page 6; Fig. S5d-f).

Fig 2 and 3: the authors state that there is cell-to-cell heterogeneity in miRNA-mediated translation repression, but I'm not sure the current set of data warrants this conclusion. I agree that the observed translation efficiency differs between cells and so does the fraction of cells that is translated. But this is the case for both the mir-21 and the mutant construct. In fact, some degree of heterogeneity is expected for any single cell measurement. The observed heterogeneity likely reflects, at least in part, heterogeneity intrinsic to translation of single mRNAs, as well as stochastic measurement noise and does not necessarily reflect heterogeneity in regulation by miRNAs. More advanced statistical analysis is required to draw such a conclusion.

We thank the Reviewer for this insightful comment. According to the Reviewer's suggestion, we re-analyzed our data. However, we could not exclude the possibility that heterogeneity reflects, at least in part, heterogeneity intrinsic to translation of single mRNAs. Therefore, we have decided to tone down our claim about cell-to-cell heterogeneity throughout the revised manuscript. Nevertheless, we would like to note that our single-cell data still provide important findings, e.g., miRNAs can stop the translation of target mRNAs completely within some cells (Fig. 2e and 2f).

Pages 8-9: the authors discuss mRNAs that are bound or unbound by RISC. However, they do not show what their detection threshold is with respect to the number of AGO molecules bound to an mRNA. So rather than 'unbound' these mRNAs may still be bound by AGO, but simply by fewer AGO molecules. I think it would be important to understand how many molecules of AGO are bound to mRNAs. Also, is it not possible that all mRNAs have an equal number of AGO molecules bound and the difference in AGO-associated fluorescence is caused by stochastic differences in labeling efficiency? Can the authors exclude this?

We thank the Reviewer for raising this important point. To investigate the detection threshold with respect to the number of AGO molecules, we made new reporter mRNAs that have a single miR-21 site or its mutant site. When we compared AGO signals on the reporters with or without a single miR-21 site, there was a small difference between them, which was statistically significant (d and e, below), suggesting that our method can detect at least a part of single AGO molecules. However, given that the difference was very small, our method would not be suitable to detect all single AGO molecules – as the Reviewer pointed out, “RISC-unbound” mRNAs may still be bound by AGO. Therefore, we have now described the limitation of our method in the revised manuscript (the Results section, page 9). Accordingly, “RISC-bound” and “RISC-unbound” were changed into “RISC-positive” and “RISC-negative”, respectively, in the revised manuscript. The results of new reporter mRNAs were added into Fig. S8g and h.

As the Reviewer pointed out, stochastic differences in labeling efficiency can affect, at least in part, the fluorescence intensity of AGO. However, while the fluorescence intensity of single SunTag molecules (SunTag signals that do not colocalize with mRNAs), which have an equal number of GCN4 epitopes (i.e., 24), showed a narrow distribution (Fig. 2h, dashed lines, more than 80% of signals distributed within the median \pm 25% range (median, \sim 100)), the fluorescence intensity of AGO on mRNAs showed a broad distribution (Fig. 3g, a blue solid line, less than 40% of signals distributed within the median \pm 25% range (median, \sim 40)). Therefore, it is unlikely that all mRNAs have an equal number of AGO molecules bound.

Page 8-9: When we used these data for single-cell analysis, RISC-binding efficiency and translational efficiency were negatively correlated (Fig. 4d), validating our experiments. Unexpectedly, single-mRNA analysis revealed that translated mRNAs tend to be bound by RISC, compared with untranslated mRNAs (Fig. 4e). These two statements appear to be contradictory. I'm having trouble understanding how these statements/findings can both be true. Better explanation would be helpful.

We agree with the Reviewer that these two statements appear to be counterintuitive at first glance. RISC prefers translated mRNAs (and requires a relatively long time to repress their translation), so RISC-bound mRNAs tend to be translated mRNAs at the single-mRNA levels. We speculate, however, since RISC-binding ultimately causes translational repression, cells

with high/low RISC-binding efficiency tend to be cells with low/high translational efficiency, respectively, at the single-cell level. We have now added this explanation in the revised manuscript (the Results section, page 11).

Figure 4e-h (and others). The reported effects are quite small in some cases and error bars are lacking in a number of graphs. Moreover, no statistics are done to support the claim of observed differences. This should be addressed.

We appreciate this suggestion. As the Reviewer pointed out, the reported effects in Fig. 4e-h are not big. Therefore, we have now provided additional data to show these differences are statistically significant (f and g, below). These results were added into the revised manuscript (Fig. S11a and b).

Section on: “Unexpectedly, single-mRNA analysis revealed that translated mRNAs tend to be bound by RISC, compared with untranslated mRNAs” as well as similar statements in the discussion: The fact that translating ribosomes stimulate binding of RISC to mRNAs was recently reported (using a similar assay) (PMID:32661421). It might be appropriate to discuss their findings within the context of this earlier work.

This citation revealed that translating ribosomes unfold mRNA structures within the ORF to unmask siRNA sites, thereby promoting interactions between target mRNAs and siRNA-

loaded AGO. We think, however, this mechanism would not explain why miRNA-loaded AGO prefers translated mRNAs because miRNA sites are generally within the 3' UTR, where ribosomes should not have access. This is why we did not cite PMID:32661421 in our original manuscript. Nonetheless, in light of the Reviewer's comment, we now think this information might be helpful to readers. Therefore, we have now discussed these points and cited the reference in the revised manuscript (the Discussion section, page 15).

Figure 5: there may be some leaky expression of the inducible transcription reporter. Therefore, some mRNAs may have been produced before ponA addition and would have thus been in the cytoplasm far longer and therefore have had more time to accumulate AGO. To control for this, the authors should include a "no PonA" control and compare the results to this control.

We understand the Reviewer's concern since some inducible promoters (e.g., the Tet-On promoter) show leaky expression. The PonA-inducible promoter, however, is known for having almost no leaky expression – this promoter exhibits basal activity that is 500-fold lower than the Tet-On system and 20-fold lower than the Tet-Off system (PMID: 8622939; PMID: 11114195). In our previous work, we confirmed that the PonA-inducible promoter had no detectable leaky expression in U2OS cells (PMID: 24069527, please see Figure 4A and 4B in the cited paper). We should have mentioned these points in the original manuscript and apologize for an insufficient explanation. We have now added an explanation and these citations for the PonA-inducible promoter in the revised manuscript (the Results section, page 12).

Throughout the figures, I find the axis labeling of RISC binding efficiency a little confusing, because efficiency is generally not expressed in a.u.. Rather, this graphs appears to show AGO immunofluorescence staining intensity I believe. A more accurate description would be preferable.

We thank the Reviewer for this helpful comment. In this study, RISC-binding efficiency was

calculated by division of the total immunofluorescence intensity of AGO on mRNAs by the number of mRNAs expressed. We have now clarified this in the Figure Legends (page 36; please also see “Data analysis” in the Methods section).

Figure 6C: This figure shows the relative abundance of the miRNA reporter mRNA rather than mRNA stability (as stated on the y-axis).

We agree with the Reviewer that “relative abundance” is more accurate. We have now changed “mRNA stability” into “relative abundance” throughout figures (Figs. 1d, 1e, 6c, and S2).

Reviewer #3:

In this manuscript entitled “Single-molecule imaging of microRNA-mediated gene silencing in cells”, Kobayashi et al. report on the development of methods to study miRNA-mediated gene silencing in situ, at single-molecule resolution. They create plasmid-based reporter systems and utilize a combination of SINAPS, single-molecule FISH and immunofluorescence (IF) to quantify nascent protein output, mRNA levels and AGO2 binding within individual cells. Using this system, they find that miRNAs preferentially bind translated/translating mRNAs, potentially right after nuclear export of mRNAs, and mediate translational repression and decay within 30 min and 60 min respectively, after mRNA binding. The authors suggest that the method provides a framework for studying spatiotemporal mRNA regulation inside cells.

The manuscript is well written. The method has been aptly described and holds immense potential to address long-standing questions on mechanisms of miRNA-mediated gene silencing. However, many (if not all) of the tools used here, except for the specific plasmid constructs, have already been reported by the authors (and others, eg – Wu et al, Science,

2016; Adivarahan et al, Mol Cell 2018 and Ruijtenberg et al, NSMB, 2020). While the method holds promise in studying mRNA regulation by other RBPs, which would be of broad interest to the scientific community, the generalizability of the technique has not been demonstrated. In addition, there still remains concerns about the ability to quantify AGO2 (or any other RBP) binding to individual mRNAs at the spatial resolution described in this manuscript, which may significantly impact extension of this method. As it stands, the manuscript is mechanistic in nature and it's unclear whether a generally applicable novel method has been developed. Below is a list of my concerns.

We appreciate the Reviewer's valuable feedback. We can understand the Reviewer's concerns about the level of advance from a methods perspective. We carefully read the comments from the Reviewer and revised our manuscript as much as possible (please see below).

1. Can the authors provide a rationale for the choice of sequences used as miR-21 sites and the corresponding negative control? A 11-mer miR21-mRNA match seems low, is that typical of miR(21)-mRNA matches? Also, typical negative controls for miRNA binding use seed mismatches (1-3nt), whereas ablation of guide strand is used here, which also tends to be drastic. In this case, wouldn't the miR21 passenger strand bind to the miR21 mismatch site? To validate miRNA site specificity, the authors should also perform assays with antimiRs.

As the Reviewer knows, RISC binds to target mRNAs through base-pairing between nucleotides 2–8 of miRNAs (seed region) and target mRNAs. In addition, base-pairing between nucleotides 13–16 (3' supplemental region) of miRNAs and target mRNAs support RISC-binding. This is why we adopted a 11-mer match, which is typical for miRNA reporters. As the Reviewer mentioned, reporter mRNAs with seed mismatches can be used as negative controls. However, recent studies revealed that RISC can bind to mRNAs only by base-pairing of the 3' region of miRNAs even when seed region does not form base-pairs (PMID: 31324449; PMID: 31806698). As such, we introduced mismatches into both the seed region and the 3' supplemental region. We can understand the Reviewer's concern that the miR-21

passenger strand seems to bind to the negative controls at first glance. However, as the 5' to 3' direction is opposite, the miR-21 passenger strand does not bind to the negative controls. Just in case, we have checked the expression level of the miR-21 passenger strand and confirmed that its expression is negligible in U2OS cells (a, below). Taken together, our reporter mRNAs would not need anti-miRs – indeed, the majority of miRNA studies does not use anti-miRs. Please note that we have already confirmed that our reporter mRNA and its negative control are actually bound by or not by RISC, respectively. In the revised manuscript, we have now clarified the rationale for the choice of sequences (the Results section, page 4; the Methods section, page 17). We have now added the data of the miR-21 passenger strand (Fig. S1b).

2. Seems like plasmids were introduced into cells by nucleofection, wouldn't that introduce heterogeneity of plasmid uptake and consequently mRNA expression? Wouldn't cells expressing very high amounts of mRNAs seemingly escape repression? Were specific cells used for analysis? If yes, the rationale has to be mentioned.

We thank the Reviewer for raising this point. In our experimental condition, nucleofection did not create large fluctuations in mRNA expression (Fig. S6a). To answer the Reviewer’s question, we have now analyzed whether cells expressing high amounts of mRNAs tend to escape from miRNA-mediated repression. However, there was no significant difference in the efficiency of translational repression between cells expressing higher and lower amounts of mRNAs (b, below). This was also true when we analyzed RISC-binding efficiency (c, below). We have now added these results in the revised manuscript (the Results section, pages 6 and 9; Figs. S6 and S8). We have not used specific cells for analysis.

3. The authors mention that the ability to measure mRNA decay, RISC binding and translation extent on a cell-by-cell basis highlights heterogeneity of miRNA-mediated gene silencing. While heterogeneity is somewhat expected, being able to assess the basis of heterogeneity is indeed powerful. Is there any specific insight the authors could provide on the potential contributor of heterogeneity? For instance, any correlations to cell-by-cell RISC concentration differences or cell cycle states will certainly underscore the importance of measuring heterogeneity.

We appreciate the Reviewer's positive comments on our methods. According to the Reviewer's suggestion, we have analyzed whether the expression level of RISC contributes to the cell-to-cell differences. These analyses, however, showed no clear correlation between them (d and e, below). We also tested whether nuclear size, which increases during the cell cycle (PMID: 20711190; PMID: 23277088), contributes to the cell-to-cell differences, but these analyses yielded similar results (f and g, below). We have now added some of these data in the revised manuscript (the Results section, pages 6 and 9; Figs. S6 and S8).

4. In fig. 4c, it is indeed intriguing to find that 48% of RISC(-) 8xmiR21 transcripts are "untranslated" whereas this population is only 32% of RISC(-) 8xmiR21 mutant transcripts. Does this imply inherent variability in translation extent, independent of miRNA mediated repression? Can the authors use this metric to arrive at a "sensitivity" of measuring miRNA-mediated gene silencing by this assay? Moreover, what is the Suntag intensity distribution of the ~10% of RISC(+) mutant transcripts? Is this similar to wild-type transcripts? Does this also impact assay "sensitivity"?

We are grateful to the Reviewer for raising this important point. The difference between the 8x miR-21 reporter and its mutated reporter is only the miRNA-binding sequence – the rest of sequences, ~5,000 nucleotides, are completely identical, so it is unlikely that these reporters have inherent variability in translation extent independent of miRNAs. To address the Reviewer's concern, we examined the sensitivity of our method – we made new reporter mRNAs that have a "single" miR-21 site or its mutated site and tested whether single RISC molecules can be detected by our method. When we analyzed RISC signals on mRNAs, the difference between the 1x miR-21 reporter and its mutant was quite small (h and i, below), suggesting that our method is not suitable for detecting single RISC molecules. Therefore, we speculate that more mRNAs are actually bound by RISC than we detect and this explains in part why there is a substantial number of RISC (-) 8x miR-21 transcripts that are untranslated. In addition, it is possible that a part of RISC (-) untranslated mRNA is already

silenced and released from RISC. In the revised manuscript, we have now added these considerations in the Results section (pages 9 and 10). The data for the new reporters are shown in Fig. S8.

Regarding ~10% of RISC (+) mutant transcripts, we would like to note that all miRNA sites of the top 30 most abundant miRNAs in U2OS cells were carefully removed from our reporter mRNAs (Fig. S1b and c) – mutant mRNAs should not be bound by RISC. Therefore, ~10% of RISC (+) mutant transcripts could be attributed to the coincidence of the 3D positions of RISC and mutant mRNAs by chance. Consistent with this, when we analyzed the SunTag intensity of both RISC (+) mutant mRNAs and RISC (+) wild-type mRNAs, RISC (+) mutant mRNAs were not translationally silenced (j, below). We should have mentioned these points in the original manuscript and apologize for an insufficient explanation. We have now added this data in Fig. S10 and clarified this point in the revised manuscript (the Results section, page 10).

5. The calculation of the number of ribosomes (as mentioned in S4) is very approximate and is perhaps not needed given the data in 2h.

We agree with the Reviewer that the calculation of the number of ribosomes is approximate, but we think the roughly-estimated number of ribosomes would still be helpful for readers compared with raw SunTag intensity. In light of the Reviewer's suggestion, we have now moved the data of the number of ribosomes to Supplementary Figures (Fig. S6).

6. It is surprising to see widespread punctate distribution of AGO, both outside and on mRNA spots. Based on AGO abundance, as reported in this manuscript, and without any super-resolution imaging, AGO signal distribution is likely to be diffuse. If the authors used any specific image processing modules to accentuate peaks for AGO IF, they should provide that info and show whether such processing alters true signal distribution. Do other proteins with similar cellular abundance have such punctate distribution? Since AGO IF is a key component of the paper, the authors should provide additional controls (2ndary antibody only and peptide blockers) for this analysis?

We appreciate the Reviewer's insightful comments. We have not used any specific image processing modules. First, our method is not robust enough to detect single AGO molecules unequivocally as described above (point #4). Second, in addition to reporter mRNAs, there should be a reasonable number of endogenous transcripts harboring multiple miRNA-binding sites, which concentrate AGO molecules. We speculate, for these reasons, AGO signals showed punctate distribution instead of diffuse distribution. To address the Reviewer's concern, we tested another AGO antibody and checked its staining pattern. This experiment confirmed that both AGO antibodies (the original mouse antibody and the additional rat antibody) showed punctate distribution (k, below), supporting the reliability of AGO IF in this study. We have now added these data in Fig. S8 (the Results section, page 8).

7. In fig. 4d, maybe its marred by limited statistics, but the negative correlation is very mild, at best one can say RISC-binding and translation efficiency are uncorrelated. There does

seem to be a trend, which may be depicted by grouping translation efficiency into high, medium and low bins and plotting RISC binding efficiency. The bins may be (semi)objectively chosen based on the distribution of Suntag intensity of the control transcript.

According to the Reviewer's suggestion, we re-analyzed our data by grouping translation efficiency into bins, such as high, medium, and low. However, the results of grouping analyses were not easy to interpret – RISC-binding efficiency actually showed statistically significant difference between bins with some ways of data grouping, but the results of statistical tests changed depending on how we grouped our data. We speculate this is because, as the Reviewer pointed out, the negative correlation is very mild. Given the results of additional analyses, we have decided to tone down our claim about the negative correlation in the revised manuscript.

8. In fig. 4, how does the translation and RISC binding trends look for the control RNA? In addition, what is the distribution of Suntag intensity of Translation +/- and RISC +/- molecules for both the control and 8xmiR21 transcripts? There may be cues in there to potentially distinguish different populations of translationally repressed transcripts and may provide hints on the step of translation that is being inhibited.

While the 8x miR-21 reporter showed a trend of mild negative correlation, the control reporter did not show such a trend (1, below). We have now added this data into Fig. 4.

In response to the second question, we re-analyzed the distribution of SunTag intensity of RISC +/- molecules for both the control and the 8x miR-21 reporter. In addition to when we analyzed RISC (+) molecules (j, above), the 8x miR-21 reporter showed lower SunTag intensity than the control even when we focused on RISC (-) molecules (m, below). This is consistent with what we mentioned in point #4: 1) some RISC-negative mRNAs would be actually bound by RISC and 2) a part of RISC-negative mRNAs would be the mRNAs already silenced and released from RISC. We have now added this data in Fig. S10.

9. The authors mention (in page 9) that “RISC does not activate translation”. It has been shown that RISC may activate translation in special scenarios (Vasudevan et al, Science, 2007). Correlations of ribosomal occupancy on mRNAs with cell cycle states (based on DAPI and cell morphology) may shed more light on such mechanisms, which is possible with the method used by the authors.

We are grateful to the Reviewer for this insightful comment. As the Reviewer pointed out, it has been reported that RISC can activate translation under serum-starved growth-arrest conditions (PMID: 18048652). In the study presented here, however, we focused on steady-state conditions, so our data would not be suitable to analyze RISC-mediated translational activation. This is why we have not cited this interesting paper in our original manuscript. Nonetheless, in light of the Reviewer’s comment, we have now cited PMID: 18048652 in the revised manuscript to provide this information to readers (the Results section, page 11).

10. In fig. 5b and 6b, is there a statistically significant retention of 8xmiR21 transcripts in the nucleus 60 min after induction, as compared to the control? Is that due to differences in constructs or related to miRNA biology?

To answer this question, we have now performed additional statistical analyses for Fig. 5b and 6b. When we compared the 8x miR-21 reporter to the control, there was no statistically significant difference at 60 min in Fig. 5b (n, below), indicating that there is no retention of 8x miR-21 reporters in the nucleus. Although the difference between the 8x miR-21 reporter and the control at 60 min was statistically significant in Fig. 6b (o, below), the 8x miR-21 reporters used in Fig. 6b are degraded by miRNA-mediated mRNA decay in the cytoplasm at 60 min, so we speculate this action of miRNAs caused the reduction in the ratio of the number of cytoplasmic mRNAs to that of nuclear mRNAs. Consistent with this, the cytoplasmic to nuclear ratio did not change at 30 min (o, below), where miRNA-mediated mRNA decay does not take place. Together, it is unlikely that the difference in Fig. 6b indicates a retention of 8x miR-21 reporters in the nucleus.

11. Prior literature suggests that AGO2 is found in the nucleus (Gagnon et al, Cell Reports, 2014). Is AGO IF signal, as measured by the authors, found in the nucleus? If yes, does AGO bind transcripts there? Did the authors analyze only cytoplasmic transcripts?

As the Reviewer pointed out, it has been suggested that AGO2 is found in the nucleus and in the cytoplasm at the comparable levels in some cell lines (PMID: 24388755). In U2OS cells, however, the majority of AGO IF signal was observed in the cytoplasm (p, below), so we focused on cytoplasmic transcripts. We have now added this data (Fig. S8) and mentioned this point in the revised manuscript (page 8).

12. Consistent with immediate export of transcripts, are RISC bound transcripts closer to the nuclear periphery at early time points? Does the spatial pattern of localization Suntag/mRNA/RISC (nuclear proximity vs cell boundary proximity) vary between the control and test constructs?

This is exactly what we would like to test next. For analyzing the spatial patterns of 1) RISC-negative untranslated mRNAs, 2) RISC-negative translated mRNAs, 3) RISC-positive untranslated mRNAs, and 4) RISC-positive translated mRNAs in a quantitative manner, we need to develop a state-of-the-art algorithm that can analyze their intracellular localization patterns depending on 3-color colocalization status. Although we will continue our best efforts to develop such an algorithm, we will pursue it as a separate study.

13. In fig. S10c, the fraction of Translation-/RISC- mRNAs seem constant (~50%) over time for the 8xmiR21 construct, whereas this fraction decreases over time for control construct (from ~60% to ~30%). What is keeping these 8xmiR21 constructs untranslated? Is there a miRNA-independent repression mechanism at play here?

We apologize for missing explanations for these results. As described in the point #4, the sequences of the 8x miR-21 reporter and its control reporter are completely identical except for the miRNA-binding sequence, so it is unlikely that the 8x miR-21 reporter selectively undergoes miRNA-independent repression – the difference in Fig. S12c should be miRNA-dependent. Please note that, as described in the point #4, 1) some RISC-negative mRNAs would be actually bound by RISC and 2) a part of RISC-negative mRNAs would be the mRNAs already silenced and released from RISC. In the revised manuscript, we have now added explanations (pages 9 and 10) so that readers can interpret the results in Fig. S12c.

14. In fig. 6c, mRNA stability seems to decrease at 30 min when comparing 8xmiR21-0 min with 8xmiR21-30 min, suggesting that the onset of decay is similar to translational repression. It is unclear as to how statistical comparisons were made, but decay is best measured by comparing mRNA levels at later time points to the earliest (0 min) for the same construct. Additional time points between 0 and 30 min will resolve the question about the temporal order of translational repression and mRNA decay.

We thank the Reviewer for this helpful comment. In Fig. 6c, mRNA stability (relative abundance of SunTag mRNAs to Fluc mRNAs) was normalized to the value of the control construct “at each time point”, so we cannot compare 8x miR-21 at 0 min with 8x miR-21 at 30 min – we can only compare 8x miR-21 with the control construct at the same time point. We should have clarified this in our original manuscript and apologize for our insufficient explanation. Please note that we performed exactly the same statistical analysis (Dunn’s multiple comparisons test) for miRNA-mediated translational repression (Fig. 5e) and for miRNA-mediated mRNA decay (Fig. 6c), so our data actually indicate that translational repression precedes mRNA decay. In the revised manuscript, we have now added

explanations for how we analyzed data and how we performed statistical tests in the Figure Legends (pages 39-40; please also see “Data analysis” and “Statistical analysis” in the Methods section).

15. The authors suggest that it takes 30 min for miRISC to repress mRNAs after binding. Do the authors suggest that an “immature” RISC binds early on, inducing a temporal order of protein binding that results in a “mature” silencing complex? The authors can test this by performing combined SINAPS+smFISH and IF for other silencing-related proteins (TNRC6, CCR4-NOT, DCPs). In addition, replacing miRNA sites for siRNA sites in the 3'UTR will also inform whether minimal RISC (AGO+si/miRNA) binds first. Here binding and cleavage can be simultaneously measured.

In this study, we just claim that RISC needs ~30 min to repress translation after the binding to mRNAs – we do not intend to claim that RISC binds to mRNAs as an immature form. We believe, however, the Reviewer’s comment raises an important question “why does RISC need ~30 min to repress translation”. In the revised manuscript, we have now mentioned that this question should be addressed in future work (the Discussion section, page 16).

16. The authors provide an explanation for why miRNAs prefer 3'UTRs – as RISC is too slow to compete with ribosomes. How does this explain active miRNA sites in 5'UTR and CDS? Is it also possible that RISC may bind such response elements prior to translation? The authors can test this by modifying the constructs, i.e. moving miRNA sites to different locations on the construct, and doing similar assays/analysis. Recent evidence (Ruijtenberg et al, NSMB, 2020) suggests that ribosomes pave the way for siRNA binding in CDS and perhaps the 3'UTR. This seems somewhat consistent with the need for RISC to bind translated/translating mRNA, but does not necessarily mean RISC cannot bind and repress CDS or 5'UTR sequences. The authors can perform abovementioned (point 15) assays to address this question.

We are grateful to the Reviewer for raising this point. RISC needs ~30 min to repress

translation after the binding to mRNAs. As this is much slower than “typical” translation initiation rates (PMID: 23791185), miRNA sites should be in the 3’ UTR in general – RISC sitting on the regions other than the 3’ UTR would be removed from mRNAs by translating ribosomes before it represses translation. This means, in other words, miRNA sites in the 5’ UTR and the CDS can still be active for the “specific” mRNAs whose translation rates are slower than ~1 per 30 min. Indeed, translation initiation rates differ depending on each mRNA species (PMID: 23791185) and are dynamically regulated in response to cellular contexts (PMID: 20094052; PMID: 27426745; PMID: 30038383). We would like to emphasize that our model does not exclude noncanonical miRNA sites in the 5’ UTR and the CDS. In light of the Reviewer’s comment, we have now added this explanation into the Discussion section of the revised manuscript (page 16). Please note that most miRNA sites are located in the 3’ UTR, but not in the 5’ UTR and the CDS (PMID: 17612493; PMID: 30638669), so we focused on miRNA sites in the 3’ UTR in this study. We hope that our methods will also contribute to understanding atypical miRNA sites in the 5’ UTR and the CDS.

17. For the colocalization analysis, why were different distance cut-offs chosen for mRNA-SunTag and mRNA-AGO colocalization analysis?

We apologize for missing explanations for the colocalization analysis. Compared with SunTag IF spots (Fig. S6b, median, ~250 spots per cell), AGO has many more IF spots (Fig. S8b, median, ~2000 spots per cell), which cause the non-specific coincidence of the 3D positions of mRNAs and AGO by chance. To minimize such false colocalizations, we adopted the shorter maximum allowed distance for mRNA-AGO colocalization analysis. We have now clarified this point in the Methods section of the revised manuscript (“Colocalization analysis” in page 23).

REVIEWER COMMENTS

Reviewer #1 (Remarks to the Author):

The manuscript "Single-molecule imaging of microRNA-mediated gene silencing in cells" by Kobayashi and Singer uses mRNA, nascent peptide and AGO labeling to analyze miRISC mediated gene regulation. Authors indicate that miRISC binds to target mRNAs after their transport from the nucleus to the cytoplasm, induces translational repression that is followed by mRNA decay and that miRISC preferred translated mRNAs over untranslated mRNAs. The revised version of the manuscript has provided additional data and control experiments, as well as improved substantially in overall explanation and discussion of the presented data. The work confirms the current model in the field (translational repression followed by mRNA decay) with additional data pointing towards selectivity of miRISC targeting (translated vs untranslated mRNAs) and would be of interest for the audience interested in miRNA mediated gene silencing and mRNA translation.

Reviewer #2 (Remarks to the Author):

In this revised version of the manuscript, the authors have nicely addressed most of my earlier comments and I support publication of the revised manuscript, after addressing two remaining points that were not quite satisfactory yet.

1. Many bar graphs still lack error bars. These are generally the results of the bulk analysis (i.e. 1 value), which is, I presume, the reason that no error bars were included. I would think, though, that for each experiment such a bulk analysis was performed and that the variation between experiments can be represented using error bars in all of these graphs.

2. Authors response: This citation revealed that translating ribosomes unfold mRNA structures within the ORF to unmask siRNA sites, thereby promoting interactions between target mRNAs and siRNA loaded AGO. We think, however, this mechanism would not explain why miRNA-loaded AGO prefers translated mRNAs because miRNA sites are generally within the 3' UTR, where ribosomes should not have access. This is why we did not cite PMID:32661421 in our original manuscript. Nonetheless, in light of the Reviewer's comment, we now think this information might be helpful to readers. Therefore, we have now discussed these points and cited the reference in the revised manuscript (the Discussion section, page 15).

Reviewer response: While it is true that PMID:32661421 examines siRNA binding sites in the ORF, this study also specifically examined binding of RISC in the 3'UTR and found that ribosomes still promote binding of RISC to its target through unfolding of mRNA secondary structure, even when the target site is located downstream of the ORF (See figure 4), as is the case in the current study. Thus, these earlier findings are relevant to the conclusions of this study, and translation of the miRNA reporter mRNA by ribosomes could explain the preferential binding of RISC to translating mRNAs in the current study. Therefore, the discussion ("In contrast to siRNA sites, however, most miRNA sites are located within the 3' UTR, where ribosomes do not access. Therefore, this mechanism for siRNAs is not relevant for

miRNAs.”) is not accurate.

Reviewed by Marvin Tanenbaum

date: 12 Oct 2021

Reviewer #3 (Remarks to the Author):

The revised version of this manuscript by Kobayashi et al has been significantly improved since its original submission. The authors have addressed most of my concerns, but a few more remain.

1. The authors are recommended to take a deeper dive into the literature for the expression of other AGOs in U2OS, since this contributes to the sensitivity of the assay and at least account for some of translation-RISC- spots. For instance, Shen et al, PLoS genetics, 2021, shows significant expression of AGO1 in U2OS cells, at least by western blotting. Since AGO2 detection is critical in this manuscript, the authors should provide secondary antibody only controls and look for expression of AGO1, 3 and 4 independently using western blotting and immunofluorescence (several commercial antibodies are available that are specific to each AGOs).

2. It is important that the authors thoroughly validate the AGO2 antibody staining. The authors can perhaps choose 2-3 cytoplasmic RNA binding proteins (RBPs) with similar expression levels as AGO2 in the Beck et al reference, and look at their staining pattern by IF. In addition to adding confidence to the spotty AGO2 pattern, the generalization of looking at other (spotty) RBPs can also be shown. The authors' argument for spotty AGO2 signal suggests that all AGOs are mRNA bound, which need not be the case. In fact, reports from other cell lines (Wang et al, Genes Dev, 2012) suggest that AGO is present at $\sim 1.5 \times 10^5$ copies per cell, which is an order of magnitude higher than reported here. Also, do some of these AGO spots colocalize with P-body components? The authors should test this. If there are ~ 15000 copies of AGO2 per U2OS cell whose volume is $\sim 4000 \mu\text{m}^3$ (bionumbers), wouldn't the signal be high enough not to resolve individual spots. If the authors suggest that each spot has more than one AGO molecule, the authors should use purified protein/antibody calibrations to precisely calculate the stoichiometry of AGO spots. Overall, a detailed characterization of the AGO2 staining is necessary.

3. The authors need to clarify several aspects of the CHX experiment. In the presence of CHX, how was translation affected overall (both the mutant and WT miR21 mRNAs), irrespective of RISC binding? CHX blocks elongation, but tends to retain ribosomes on messages, so CHX treated cells will have mRNAs with nascent peptides. If translation and RISC binding are related, isn't it expected that AGO signals will increase with CHX treatment? Are the authors suggesting that translation elongation is a prerequisite for RISC binding?

Point-by-point response to the Reviewers' comments

Reviewer #1:

The manuscript "Single-molecule imaging of microRNA-mediated gene silencing in cells" by Kobayashi and Singer uses mRNA, nascent peptide and AGO labeling to analyze miRISC mediated gene regulation. Authors indicate that miRISC binds to target mRNAs after their transport from the nucleus to the cytoplasm, induces translational repression that is followed by mRNA decay and that miRISC preferred translated mRNAs over untranslated mRNAs. The revised version of the manuscript has provided additional data and control experiments, as well as improved substantially in overall explanation and discussion of the presented data. The work confirms the current model in the field (translational repression followed by mRNA decay) with additional data pointing towards selectivity of miRISC targeting (translated vs untranslated mRNAs) and would be of interest for the audience interested in miRNA mediated gene silencing and mRNA translation.

We appreciate the Reviewer's positive comments for publication.

Reviewer #2:

In this revised version of the manuscript, the authors have nicely addressed most of my earlier comments and I support publication of the revised manuscript, after addressing two remaining points that were not quite satisfactory yet.

1. Many bar graphs still lack error bars. These are generally the results of the bulk analysis (i.e. 1 value), which is, I presume, the reason that no error bars were included. I would think, though, that for each experiment such a bulk analysis was performed and that the variation between experiments can be represented using error bars in all of these graphs.

We are grateful to the Reviewer for positive comments and valuable feedback. Although no error bars were included in Figures 1d, 2d, 3c, 4e, 4g, 5g, and 5h, please note that the variation between experiments has already shown for Figures 4e, 4g, 5g, and 5h (Supplementary Figure 11a and 11b). Also, the purpose of Figures 1d, 2d, and 3c is to confirm the current model in the field, so we would think that error bars are not necessarily required for these data. Instead, to address the Reviewer’s concern, we have attached the results of two independent experiments for these Figures (please see below) – bulk analyses in this study have minimal variation between experiments.

2. Authors response: This citation revealed that translating ribosomes unfold mRNA

structures within the ORF to unmask siRNA sites, thereby promoting interactions between target mRNAs and siRNA loaded AGO. We think, however, this mechanism would not explain why miRNA-loaded AGO prefers translated mRNAs because miRNA sites are generally within the 3' UTR, where ribosomes should not have access. This is why we did not cite PMID:32661421 in our original manuscript. Nonetheless, in light of the Reviewer's comment, we now think this information might be helpful to readers. Therefore, we have now discussed these points and cited the reference in the revised manuscript (the Discussion section, page 15).

Reviewer response: While it is true that PMID:32661421 examines siRNA binding sites in the ORF, this study also specifically examined binding of RISC in the 3'UTR and found that ribosomes still promote binding of RISC to its target through unfolding of mRNA secondary structure, even when the target site is located downstream of the ORF (See figure 4), as is the case in the current study. Thus, these earlier findings are relevant to the conclusions of this study, and translation of the miRNA reporter mRNA by ribosomes could explain the preferential binding of RISC to translating mRNAs in the current study. Therefore, the discussion ("In contrast to siRNA sites, however, most miRNA sites are located within the 3' UTR, where ribosomes do not access. Therefore, this mechanism for siRNAs is not relevant for miRNAs.") is not accurate.

We apologize for an insufficient explanation about this point. While PMID: 32661421 revealed that translating ribosomes unfold flanking mRNA structures, thereby promoting RISC-binding even when the siRNA-binding site is located downstream of the stop codon (the figure 4 in PMID: 32661421), this was dependent on the distance between the stop codon and the binding site (the figure 5 in PMID: 32661421) – proximal siRNA-binding sites located 27 nt or 110 nt downstream of the stop codon were sensitive to translation, but the distal binding site located 542 nt downstream of the stop codon was comparable to the negative control. As our reporters have a ~700-nt gap between the stop codon and the miRNA-binding site, we initially thought that the mechanism reported in PMID: 32661421 would not explain what we found. However, given that base-pairing within mRNAs can

occur over large distances (PMID: 27180905), we now agree with the Reviewer that the mechanism for siRNAs might explain why miRNAs prefer translated mRNAs. Therefore, we have modified our manuscript accordingly (the Discussion section, page 15). We appreciate the Reviewer's feedback, which helped us to improve the manuscript.

Reviewer #3:

The revised version of this manuscript by Kobayashi et al has been significantly improved since its original submission. The authors have addressed most of my concerns, but a few more remain.

We appreciate the Reviewer's positive comments that our manuscript has been significantly improved.

1. The authors are recommended to take a deeper dive into the literature for the expression of other AGOs in U2OS, since this contributes to the sensitivity of the assay and at least account for some of translation-RISC- spots. For instance, Shen et al, PLoS genetics, 2021, shows significant expression of AGO1 in U2OS cells, at least by western blotting. Since AGO2 detection is critical in this manuscript, the authors should provide secondary antibody only controls and look for expression of AGO1, 3 and 4 independently using western blotting and immunofluorescence (several commercial antibodies are available that are specific to each AGOs).

We would like to emphasize that PMID: 22068332 does not mean that neither AGO1, AGO3, nor AGO4 are expressed in U2OS cells. Proteome analyses are generally not robust enough to detect proteins with low expression levels, so it is not surprising that AGO1 was detected in U2OS cells by western blotting (PMID: 33600493). With respect to the proteome data from PMID: 22068332, the point is that AGO2 is predominantly expressed compared with

other AGOs in U2OS cells.

To address the Reviewer's suggestions as much as possible, we purchased commercially available antibodies against AGO1, AGO3, or AGO4. Unfortunately, however, these antibodies did not specifically recognize their target AGOs even though antibody companies claim that they are specific to each AGO (please see below) – in fact, AGO1, AGO3, and AGO4 have been less studied in contrast to AGO2. Even though we could not investigate other AGOs independently, as AGO2 is predominantly expressed in U2OS cells, which means the expression of other AGOs are negligible, we would think that focusing on AGO2 is relevant in this study.

2. It is important that the authors thoroughly validate the AGO2 antibody staining. The authors can perhaps choose 2-3 cytoplasmic RNA binding proteins (RBPs) with similar expression levels as AGO2 in the Beck et al reference, and look at their staining pattern by IF. In addition to adding confidence to the spotty AGO2 pattern, the generalization of looking at other (spotty) RBPs can also be shown. The authors' argument for spotty AGO2 signal suggests that all AGOs are mRNA bound, which need not be the case. In fact, reports from other cell lines (Wang et al, Genes Dev, 2012) suggest that AGO is present at $\sim 1.5 \times 10^5$ copies per cell, which is an order of magnitude higher than reported here. Also, do some of these AGO spots colocalize with P-body components? The authors should test this. If there are ~ 15000 copies of AGO2 per U2OS cell whose volume is $\sim 4000 \mu\text{m}^3$ (bionumbers),

wouldn't the signal be high enough not to resolve individual spots. If the authors suggest that each spot has more than one AGO molecule, the authors should use purified protein/antibody calibrations to precisely calculate the stoichiometry of AGO spots. Overall, a detailed characterization of the AGO2 staining is necessary.

We apologize for an insufficient explanation about the rationale for AGO2 immunostaining. As AGO2 has been extensively studied for decades, various tools for AGO2 were already well-validated. Among them, the mouse monoclonal anti-AGO2 antibody (clone 4G8) developed by Mikiko C. Siomi (The University of Tokyo) and Haruhiko Siomi (Keio University) (PMID: 18524951; PMID: 18369776) is one of the best-characterized anti-AGO2 antibodies in the field. For example, in response to the Reviewer's question, it has already been shown that the immunostaining signals of this antibody colocalize with P-body components (PMID: 18524951). Nonetheless, to further validate this antibody, 1) we have now confirmed by ourselves that this antibody specifically recognizes AGO2, but not other AGOs (please see below, the left side), in contrast to the non-specific AGO antibodies described above (point #1). Moreover, 2) we have also confirmed in our hands that AGO2 immunostaining with this antibody actually labels P-bodies (please see below, the right side, white arrow heads) as well as punctate spots throughout the cytoplasm. Importantly, it has long been known that endogenous AGO2 shows this staining pattern, i.e., punctate spots throughout the cytoplasm + a few P-bodies (typical images can be seen in the figure 1 in PMID: 17116888) – we should have referred to this in our original manuscript and apologize for an insufficient explanation.

In this study, according to the previous comment from the Reviewer, we have also tested the rat monoclonal anti-AGO2 antibody (clone 11A9) developed by Gunter Meister (University of Regensburg) (PMID: 18430891) – this is also the well-characterized antibody in the field. Consistent with our data, it has been reported that immunostaining with the 11A9 antibody also labels punctate spots throughout the cytoplasm (e.g., PMID: 23511973, “Ago2 staining showed the typical diffuse punctate pattern throughout the cytoplasm” in the second paragraph in the Results section) – a previous study confirmed that these signals disappeared upon depletion of endogenous AGO2 (the figure 4 in PMID: 18430891). We believe that the fact that both the two established antibodies (clones 4G8 and 11A9) generated from two distinct species (mice and rats) showed the same IF pattern (Supplementary Figure 8) further supports the reliability of AGO2 immunostaining in this study. Also, we would be grateful if the Reviewer could appreciate again that the AGO2 immunostaining signals in our method actually increased on the miR-21 reporter, but not on the control reporter (Figures 3, 4, and 5).

Given the Reviewer’s comment, we have now clarified these points in our manuscript so that readers can easily acknowledge the reliability of AGO2 immunostaining (the Results section, page 8). In addition, to address the Reviewer’s concern, we have significantly toned down our claim about the generality of this study to other RBPs throughout the manuscript (the Abstract section, page 2; the Discussion section, page 16). We hope that additional data, information, and explanations above, which further support the reliability of AGO2 immunostaining, will satisfy the Reviewer’s concern.

3. The authors need to clarify several aspects of the CHX experiment. In the presence of CHX, how was translation affected overall (both the mutant and WT miR21 mRNAs), irrespective of RISC binding? CHX blocks elongation, but tends to retain ribosomes on messages, so CHX treated cells will have mRNAs with nascent peptides. If translation and RISC binding are related, isn't it expected that AGO signals will increase with CHX treatment? Are the authors suggesting that translation elongation is a prerequisite for RISC binding?

As suggested by the Reviewer, we analyzed SunTag signals for both the miR-21 reporter and the control reporter in the presence or absence of CHX. Consistent with the fact that CHX blocks elongation, SunTag signals on mRNAs were comparable between CHX- and CHX+ (please see below). Therefore, translating ribosomes might be important for RISC-binding. We have now added the data of SunTag signals into Supplementary Figure 11.

REVIEWERS' COMMENTS

Reviewer #3 (Remarks to the Author):

The authors have addressed all my concerns and I support publication of this manuscript. I recommend that the authors include important technical information about the specificities of all the AGO antibodies tested (western blotting and IF images of potential P-body staining), in the manuscript. This is valuable information for the community. Overall, this manuscript stands to address key mechanistic aspects of miRNA-mediated gene silencing.

Point-by-point response to the Reviewers' comments

Reviewer #3:

The authors have addressed all my concerns and I support publication of this manuscript. I recommend that the authors include important technical information about the specificities of all the AGO antibodies tested (western blotting and IF images of potential P-body staining), in the manuscript. This is valuable information for the community. Overall, this manuscript stands to address key mechanistic aspects of miRNA-mediated gene silencing.

We are grateful to the Reviewer for positive comments for publication. The antibodies tested in the previous point-by-point response are: anti-AGO1 mouse antibody (FUJIFILM Wako Pure Chemical, 015-22411, clone 2A7), anti-AGO2 mouse antibody (FUJIFILM Wako Pure Chemical, 015-22031, clone 4G8), anti-AGO3 mouse antibody (FUJIFILM Wako Pure Chemical, 010-23821, clone 6-107), and anti-AGO4 mouse antibody (FUJIFILM Wako Pure Chemical, 012-24741, clone 2B2).